# Implicating the red body of *Nannochloropsis* in forming the recalcitrant cell wall polymer algaenan

Christopher W. Gee [1,2], Johan Andersen-Ranberg [3], Ethan Boynton [1,2], Rachel Z. Rosen[4], Danielle Jorgens[5], Patricia Grob [1,6], Hoi-Ying N. Holman [5] & Krishna K. Niyogi [1,2,7] ✉

Stramenopile algae contribute significantly to global primary productivity, and one class, Eustigmatophyceae, is increasingly studied for applications in high-value lipid production. Yet much about their basic biology remains unknown, including the nature of an enigmatic, pigmented globule found in vegetative cells. Here, we present an in-depth examination of this "red body," focusing on *Nannochloropsis oceanica*. During the cell cycle, the red body forms adjacent to the plastid, but unexpectedly it is secreted and released with the auto-sporangial wall following cell division. Shed red bodies contain antioxidant ketocarotenoids, and overexpression of a beta-carotene ketolase results in enlarged red bodies. Infrared spectroscopy indicates long-chain, aliphatic lipids in shed red bodies and cell walls, and UHPLC-HRMS detects a C32 alkyl diol, a potential precursor of algaenan, a recalcitrant cell wall polymer. We propose that the red body transports algaenan precursors from plastid to apoplast to be incorporated into daughter cell walls.

The class Eustigmatophyceae is a clade of stramenopile microalgae that has been the subject of a growing body of work in the past decade, due in part to their rapid growth and ability to partition large fractions of their biomass into valuable lipids such as the omega-3 fatty acid, eicosapentaenoic acid (EPA)[1–3]. Algae from this class were reputed to be uncommon, but recent phylogenetic reassessments have yielded a much-expanded tree[4–6], and eustigmatophyte algae have been isolated from freshwater, marine, and terrestrial environments around the world[7–10].

In recent years, the marine nanophytoplankton *Nannochloropsis* and its sister genus *Microchloropsis*[11] have been established as model systems. Cells are typically solitary, non-motile coccoids 2–4 μm in diameter[12], and they reproduce by asexual fission during the night period[13,14]. Yields for *Nannochloropsis* lipid content have been reported up to ~50% of the biomass total[15], and much research has been directed

towards developing elite lipid-producing strains[16–18] and optimizing culture conditions[19,20]. Molecular genetic studies of these genera have been facilitated by the publication of reference genomes[21–23] and the development of gene editing tools[24–27]. Additionally, basic research has highlighted the potential of these algae for providing insights into algal $CO_2$-concentrating mechanisms[28,29], photosystem I supercomplex structure[30], light-harvesting pigments (i.e., violaxanthin and vaucheriaxanthin)[31], non-photochemical quenching[32,33], cell division[34], and endosymbiosis with a novel clade of *Phycorickettsia*[35–37].

A diagnostic characteristic of eustigmatophyte vegetative cells is the presence of a red-orange globule that resides outside of the chloroplast[7]. This feature has been observed in several eustigmatophytes[4,8,38,39], including *Nannochloropsis* specifically[5,12,40,41]. The structure has been variously referred to as a "lipid body," "reddish globule," "pigmented spherule," "eyespot," "red body," etc. The

[1]Howard Hughes Medical Institute, University of California, Berkeley, CA 94720, USA. [2]Department of Plant and Microbial Biology, University of California, Berkeley, CA 94720, USA. [3]University of Copenhagen, Department of Plant and Environmental Sciences, Frederiksberg DK-1871, Denmark. [4]Department of Chemistry, University of California, Berkeley, CA 94702, USA. [5]Electron Microscope Laboratory, University of California, Berkeley, CA 94720, USA. [6]California Institute of Quantitative Biosciences, University of California, Berkeley, CA 94720, USA. [7]Molecular Biophysics and Integrated Bioimaging Division, Lawrence Berkeley National Laboratory, Berkeley, CA 94720, USA. ✉e-mail: niyogi@berkeley.edu

original description of the class by Hibberd & Leedale established Eustigmatophyceae by extricating species from Xanthophyceae based on zoospore morphology that included an unusual extraplastidial anterior eyespot, which we presume to be the eponymous "stigma[42]." Hibberd later authored an update on the classifications of Eustigmatophyceae and included a description of a "reddish globule" in vegetative cells that was different from the "red extraplastidial eyespot" in zoospores[43]. Given the relatively common use of "red body" in existing work to refer to the red globular feature in vegetative cells, and that this appears to be distinct from the zoospore eyespot, we will refer to it here as the "red body."

Despite the apparently widespread occurrence of the red body across Eustigmatophyceae, no in-depth examinations have been performed on it beyond a simple description of size and color from microscopy. To our knowledge, no detailed account of its formation has yet been reported, nor a function proposed.

Here we present an in-depth examination of the eustigmatophyte red body. We focused on the model species, *Nannochloropsis oceanica*, and characterized the development of the red body over the course of the cell cycle by light and electron microscopy. The red body was secreted during cell division, and shed into the medium along with the autosporangial wall, which facilitated its isolation and enrichment for subsequent mass and infrared spectroscopy and chromatography. These analyses indicated the presence of ketocarotenoids and long-chain aliphatic lipids. We propose that the red body accumulates lipidic precursors of the polymer algaenan, along with ancillary proteins, and upon autospore maturation, delivers these molecules into the apoplast where they polymerize to form the recalcitrant outer

layer of the daughter cell walls. Implications for future research on eustigmatophyte algae and understanding the biosynthesis of hydrophobic biopolymers, in general, are discussed.

## Results

### The red body of eustigmatophyte algae is an autofluorescent, globular, membrane-bound organelle

In our previous studies of *N. oceanica* CCMP1779[28], we observed unexpected autofluorescent punctae that were distinct from the plastid. Although the physical origin of the emitted light from these punctae remains unknown, we will refer to it as autofluorescence. By transmitted light microscopy, a corresponding reddish globule was sometimes visible. We obtained other eustigmatophyte species and again observed punctate, extra-plastidic fluorescence that corresponded with a red body (Fig. 1). Our observation of autofluorescence from the red body is consistent with micrographs of a different eustigmatophyte, which was shown to be capable of using far-red light for photosynthesis[44]. In the species we examined, the red body autofluorescence was visible through filter sets spanning most of the visible spectrum (Supplementary Fig. 1). Additionally, for *Chloridella neglecta* (SAG 48.84), confocal scanning laser microscopy revealed that the red body was an aggregation of smaller bodies in this species (Supplementary Fig. 2). Green autofluorescence (GAF) from the red body was generally robust and spectrally distinct from red chlorophyll fluorescence, so subsequent microscopy used settings similar to those for green fluorescent protein (excitation 488 nm, emission 495–550 nm). The well-established eustigmatophyte model organism, *N. oceanica*, was selected for subsequent experiments.

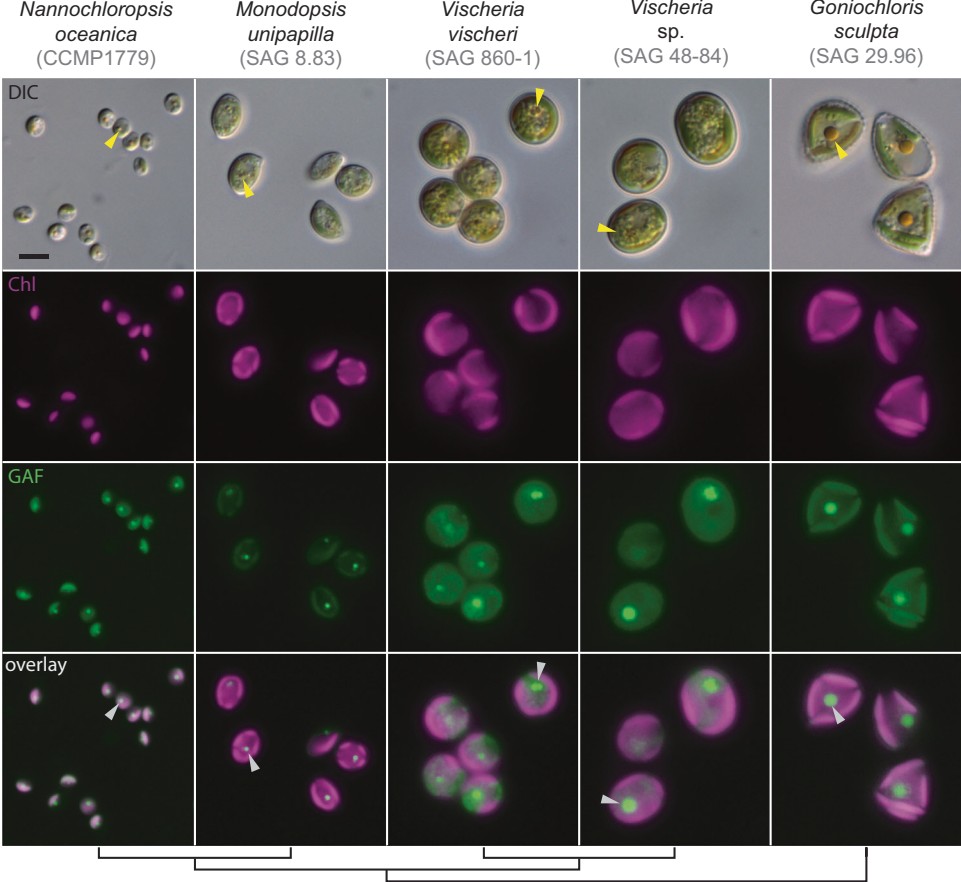

**Fig. 1 | The red body is a globular, autofluorescent subcellular structure found in eustigmatophyte algae.** Widefield fluorescence microscopy. From top row to bottom: differential interference contrast (DIC), chlorophyll autofluorescence (Chl-pseudo-colored magenta, ex/em: 660 nm/700 nm), green autofluorescence (GAF-pseudo-colored green, ex/em: 470 nm/525 nm), overlay of Chl and GAF. An example red body is indicated for each species by an arrowhead (yellow for DIC images, white for overlay). The cladogram at the bottom is based on ref. 4. The scale bar in the upper left image equals 5 µm and applies to all images in this figure.

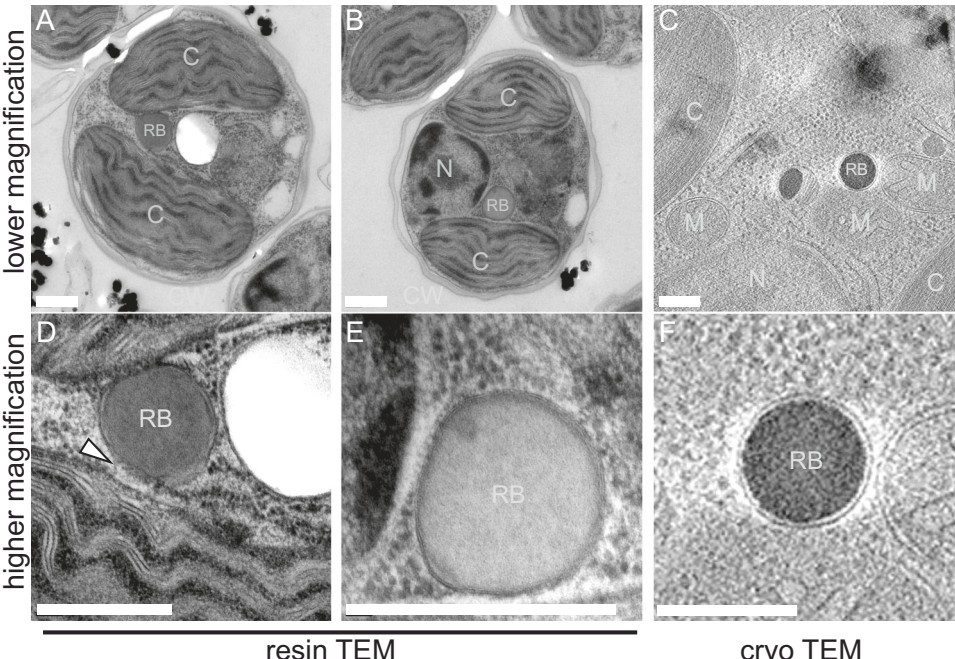

**Fig. 2 | A membrane envelopes the red body. A**, **D** + **B**, **E** Resin-embedded, heavy metal stained transmission electron microscopy (TEM) of two separate *N. oceanica* cells preserved by high-pressure freezing at subjective dusk. Presumptive assignments: CW = cell wall, C = chloroplast, RB = red body, N = nucleus. (**D**) and (**E**) show the same cells as (**A**) and (**B**), respectively, but were acquired at higher magnification. Chloroplasts are identified by stacked groups of photosynthetic thylakoid membranes that converge at the poles of the organelle, and a possible membrane connecting the red body with the chloroplast is indicated with a white arrowhead in (**D**). **C**, **F** A selected slice from the 3D cryo-tomogram capturing the red body with the apparent enveloping membrane (whole series available as Supplementary Movie 1, and details of CLEM/cryoET pipeline in Supplementary Figs. 3 and 4). Both membranes and denser carotenoid contents appear black in the cryo EM density. M = mitochondria. (**F**) is a digitally enlarged portion of (**C**) to make the enveloping membrane more visible, particularly for comparison with the nearby mitochondrial membranes. For all images, the scale bar = 500 nm.

Transmission electron microscopy (TEM) revealed additional details of red body morphology. Chloroplasts served as landmarks and were readily identified by thylakoids grouped in stacks of three, which is typical of secondary plastids derived from red algae[45]. Based on our observations by fluorescence microscopy, we identified candidate structures in TEM of resin thin sections that appeared likely to be the red body based on size, shape, and relative position to the chloroplast. These candidate red bodies were circular, homogenous and moderately electron-dense, and often located adjacent to the chloroplast (Fig. 2A, B, D, E). The maximum observed diameter was ~300–400 nm in diameter, although smaller ones were observed and will be described in later experiments. At least one membrane was clearly visible encompassing the red body (Fig. 2), and in some cases, this was enclosed by another membrane encircling the chloroplast (Fig. 2A and D). This enclosing membrane likely corresponds to the outer membrane of the chloroplast-endoplasmic reticulum (CER), a contiguous membrane network formed by the outer nuclear envelope, ER, and the outermost plastid membrane[34,46].

To complement observations made by traditional resin-embedded TEM, cryo-electron tomography was employed to investigate the native-state ultrastructure of the red body within *Nannochloropsis* and confirm our initial observations at higher resolution in three dimensions. In order to observe organelles within *Nannochloropsis* with high-resolution cryo-TEM, cryo-milling was needed to create a thinned window (cryo-lamella) inside of the 4 μm thick cell. Further, cryo-fluorescence microscopy was used to guide cryo-focused ion beam (cryo-FIB) milling of thin lamellae for cryo-electron tomography (Supplementary Figs. 3 and 4). The 350 nm thick reconstructed cryo-tomograhic volume (Supplementary Movie 1) shows the red body in high resolution within its cellular context. High densities such as membranes and carotenoids appear black on the light gray background. The tomographic volume (Supplementary

Movie 1) contained a rich collection of cellular features. These include a spherical red body ~250 nm in diameter, clearly enveloped by a membrane (still images shown in Fig. 2C and F). The membrane detection software package TomoSegMemTV used to model the 3D tomographic density confirmed the presence of the membrane and was used to label adjacent cellular compartments as described in Supplementary Fig. 4. Additionally, a great many other cellular features were visible, and the preserved detail of these structures highlights the promise of this technique for future investigations of *Nannochloropsis* cell biology. Given the clear compartmentalization and differentiation of its contents from the rest of the cell, we propose that the red body be defined as a membrane-bound organelle.

### Development of the red body is integrated into the cell cycle of *N. oceanica*

In cells grown under continuous light, the size and position of the red body were variable. Growing cells under a diurnal photoperiod (12 h light, 12 h dark) not only induced synchronous cell division but also revealed that the red body exhibits regular changes in size and position over the course of the cell cycle. *N. oceanica* cells enlarge during the day period, divide, and then separate into daughter autospores during the night period[13] (Fig. 3A). To improve the resolution of these small features, we used super-resolution structured illumination (SIM) fluorescence microscopy at timepoints throughout the day/night cycle. During the day period, chloroplasts (red fluorescence) grew from flattened ellipsoids to bi-lobed shapes which divided around the light-dark transition to form two daughter plastids (Fig. 3B, C). These subsequently divided again to form four plastids, presumably held within four daughter autospores, indicating that *Nannochloropsis* can divide by multiple fission as seen in some other algae[47]. The red bodies (green fluorescence) began to be clearly visible by ~4 h into the light period as small, round punctae next to the plastid. These continued to

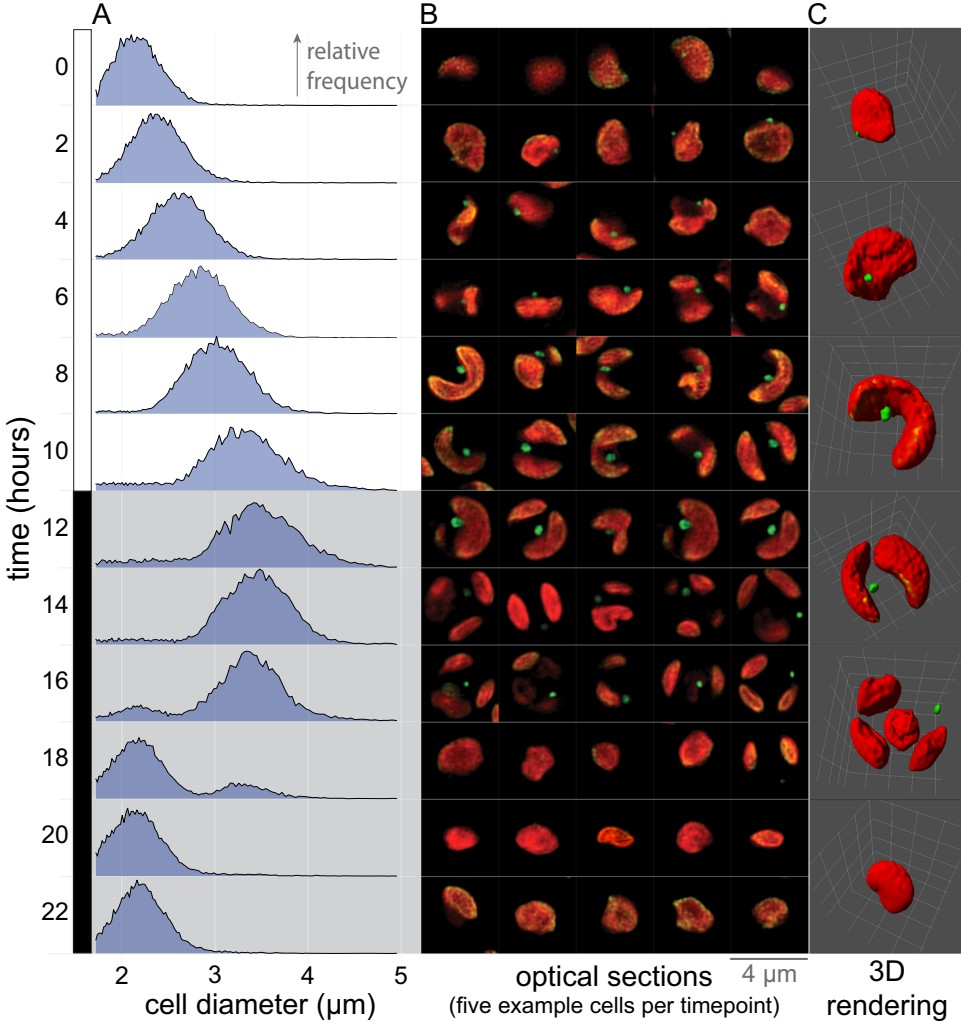

**Fig. 3 | The formation of the red body is integrated into the diurnal cell cycle of** ***Nannochloropsis.*** **A** A liquid culture of synchronously dividing cells was sampled at the indicated time points (0 hr = lights on), and cell diameter was quantified by Coulter counter. The resulting distributions are plotted as the relative frequency with the same vertical scaling (bin size ~ 0.01 μm, maximum vertical height ~3.6%, for each distribution, $n > 10k$). **B** SIM optical z-sections of cells sampled at the same time as in (**A**). Chlorophyll autofluorescence (excitation 642 nm, emission >655 nm,

pseudo-colored red); green autofluorescence (excitation 488 nm, emission 495 to 550 nm, pseudo-colored green). The Z-position for each cell was chosen to maximize the apparent diameter of the red body. Each individual cell image frame measures $4 \times 4$ μm. **C** 3D reconstructions derived from SIM z-stacks of cells chosen to represent stages of the cell cycle. Each 3D rendering is bounded by a $4 \times 4 \times 4$ μm volume, demarcated on the edges of the volume by a white grid.

grow adjacent to the chloroplast but later moved to the cell periphery during the night (Fig. 3B, C). Newly separated autospores contained one plastid but no visible red body, suggesting that it is "shed" upon autospore separation and is subsequently generated anew in each daughter cell.

### The red body is secreted into the apoplastic space prior to autospore release

To test this idea, we immobilized synchronized cells onto a polylysine-coated microscope cover glass and observed what was left adhered to the glass after cell division and separation. Initial attempts at live-cell imaging over the entire timeframe were unsuccessful, possibly due to cellular sensitivity to light during division, so pairs of coverglasses were made from the same culture and imaged before or after autospore separation (Supplementary Fig. 5A, B). Green autofluorescent punctae were observed adhered to the glass after autospore separation, sometimes trapped within a transparent, pointed, tubular structure (Supplementary Fig. 5B). We propose that these structures are shed autosporangial walls that previously contained the

growing autospores during maturation. The released autospores were then bound onto new coverglasses, and no red body punctae were observed (Supplementary Fig. 5C), again indicating that the structures are formed de novo in each daughter cell.

The cell wall of *Nannochloropsis* has been found to be made of two main layers, a thick cellulose inner layer, and a thinner, outer layer made of a hydrophobic, chemically recalcitrant polymer called algaenan[48]. In the late stages of cell division, we directly visualized the red body outside of developing autospore cell walls, but enclosed by the enveloping autosporangial wall by labeling cellulose with the fluorescent dye, calcofluor white (CFW) (Supplementary Fig. 6). We additionally observed shed red bodies within CFW-labeled shed autosporangial walls, which suggests that these shed walls retain at least some cellulose from the inner cell wall layer (Supplementary Fig. 6).

### Cells exhibit a period of increased permeability to exogenous dyes that coincides with autospore release from the enclosing autosporangial wall

In the CFW staining experiment above, relatively few cells appeared to be labeled. Low efficiency in stain uptake has been reported for this

species and close relatives[49–51], and in other algae that produce algaenan[52,53]. Algaenan cell coverings may serve as protection for resting spores against desiccation[54,55] or as defense against biotic stress[52], and it may thus form a physical barrier to exogenous dye entry.

We hypothesized that there may exist a window of increased permeability as the autosporangial wall begins to split in preparation for daughter cell release, but before the new autospore cell walls have fully matured. We thus incubated synchronously dividing cells with CFW or the DNA-binding dye, Hoechst 33342, for 3-hour periods starting at different points during subjective night, washing cell aliquots to reduce background unbound dye, and observing fluorescence of these cells in bulk culture and by microscopy (schematic in Fig. 4A; synchrony check in Supplementary

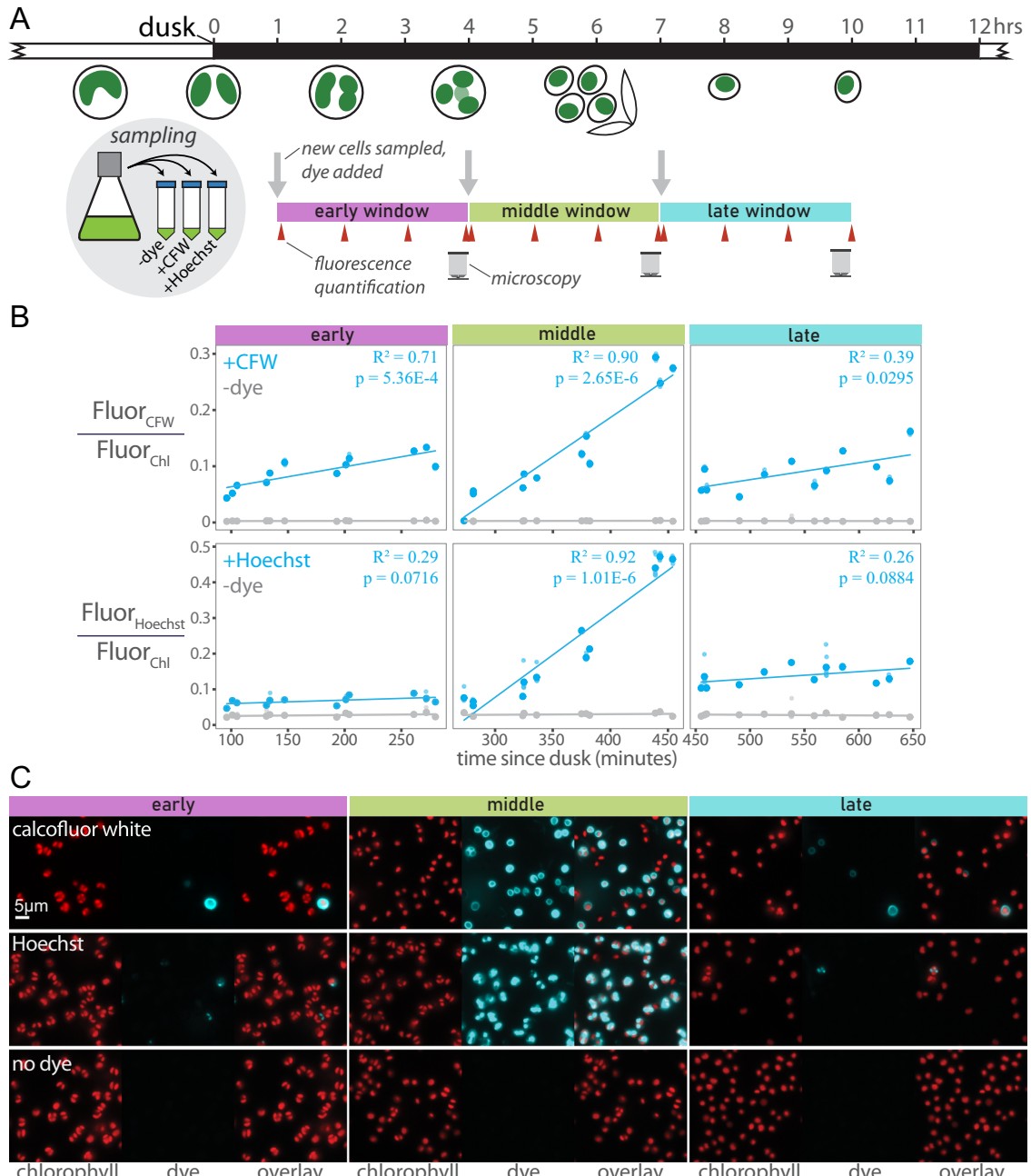

**Fig. 4 | Cells exhibit increased permeability to exogenous dyes during the period of autospore maturation and release. A** Synchronized cells were sampled from a culture at the indicated times after dusk. The sampled culture was divided into three treatments: no dye addition, calcofluor white (CFW), or Hoechst 33342. These treatments were subsampled every hour, washed of excess dye, and fluorescence measured on a plate reader (red arrowheads). At the end of the 3 h window, washed cells were imaged; CFW and Hoechst fluorescence pseudo-colored as cyan, chlorophyll fluorescence as red. These procedures were repeated with new samples from the original culture two more times. The entire experiment was carried out three times on different days. **B** Fluorescence over time from CFW or Hoechst was normalized to chlorophyll fluorescence. The three experimental runs were plotted together, and each data marker is the mean of 7 technical replicates (individual wells), which are represented by smaller data markers that generally lie inside the larger markers. $n = 3$ biologically separate trials with 7 technical replicates for each stain/timepoint combination. Interaction between fluorescence and time determined via one-way ANCOVA. Simple linear regression lines, coefficients of determination, and overall $p$-values are shown for each treatment in each panel. **C** Widefield fluorescence microscopy. For each window, chlorophyll fluorescence (red), dye fluorescence (cyan), and an overlay are shown. The scale bar in the upper left image = 5 µm, and applies to all microscopy images.

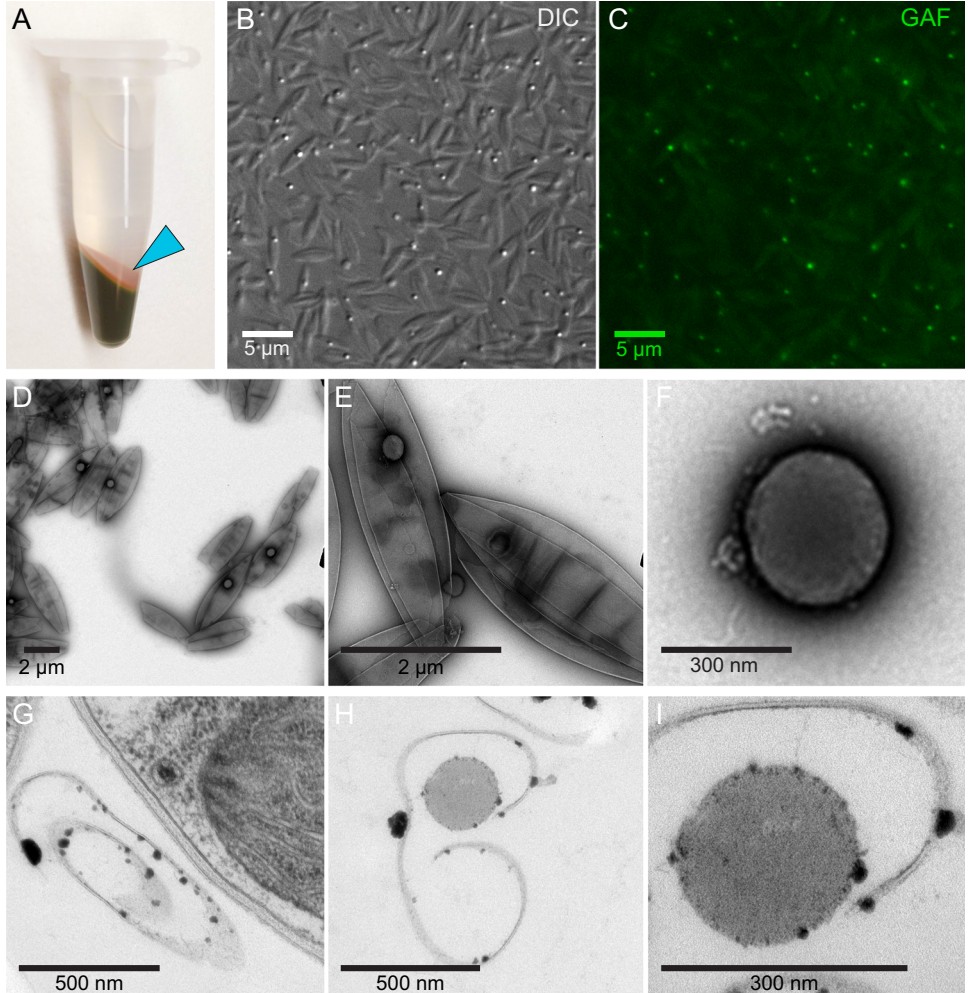

**Fig. 5 | Shed cell walls and red bodies can be collected from the media. A** 100 mL of culture was concentrated into 1 mL by centrifugation, showing the red sediment on top of the cell pellet (cyan arrowhead). **B, C** This sediment imaged by fluorescence microscopy shows elongated, pointed particles (presumably shed cell walls) and smaller, autofluorescent spherical particles (presumably shed red bodies). Differential interference contrast (DIC) transmitted light and green autofluorescence (GAF) acquired as described previously. **D, E, F** Whole mount, negative stained, transmission electron microscopy of the red sediment, showing the spherical shed red bodies wrapped in the shed cell wall (**D, E**) or alone (**F**). **G, H, I** Resin-embedded, thin section samples of concentrated culture showing an elongate particle resembling the outer algaenan layer of a nearby intact cell (**G**), an apparent cross-section of a shed wall and putative red body (**H**), and this same particle imaged at higher magnification showing a relatively homogenous interior of moderate electron density (**I**).

Fig. 7). Cells incubated with the dyes during the period of autospore release (dusk +3 to +7 h) exhibited greater rates of fluorescence increase (dye uptake) compared with time windows a few hours before or after (Fig. 4B, C). That is, the slopes of a line of best fit were different between windows (Fig. 4B). An analysis of covariance (ANCOVA) with the model Fluorescence ~ Time * Window yielded significant interaction effects between time and staining window for the CFW treatment ($p$-value = 1.66e-7), Hoechst ($p$-value = 1.44e-12), but not for the no-stain control ($p$-value = 0.245). This indicates that the slopes (rate of dye uptake or staining frequency) are different between the staining windows.

Microscopy of stained cells revealed circular outlines (cell walls) for the CFW-dyed cells and multiple punctae (nuclei) in each cell for the Hoechst-dyed cells, consistent with their expected binding targets (Fig. 4C, additional microscopy in Supplementary Fig. 8). The frequency of staining was relatively low in the early and late windows compared to the middle window (Fig. 4C, Supplementary Fig. 8). Together, these observations are consistent with a gap in algaenan coverage between the initial splitting of the maternal autosporangial wall and the formation of new algaenan layers around autospores.

## Shed cell walls and red bodies contain ketocarotenoids and a long-chain alkyl diol

We observed a "red sediment" on top of cell pellets of *Nannochloropsis* when harvesting large numbers of cells by centrifugation (Fig. 5A). This extracellular debris has also been reported in a study of *Nannochloropsis* media recycling, and a study on the structure of the cell wall[48,56]. This red sediment consisted of particles matching the appearance of shed autosporangial walls and red bodies (Supplementary Fig. 5, Fig. 5B and C). TEM revealed the rolled and wrinkled appearance of the autosporangial walls (Fig. 5D, E, and F) that were similar in thickness (~5 nm) to the outer, algaenan cell wall layer of *Nannochloropsis* (Fig. 5G, H, and I)[34,48]. The shed red bodies appeared as roughly spherical and homogenous particles slightly less than ~300 nm in diameter (Fig. 5F and I).

Daily harvesting of dense, actively growing cultures produced relatively large amounts of the red sediment, and high-pressure homogenization selectively disrupted the shed autosporangial walls, allowing us to obtain enrichments of shed red bodies (Fig. 6A, Supplementary Fig. 9). Organic solvent extracts of the red sediment and red body enrichment were subjected to ultra high-pressure liquid chromatography coupled to high-resolution mass spectrometry

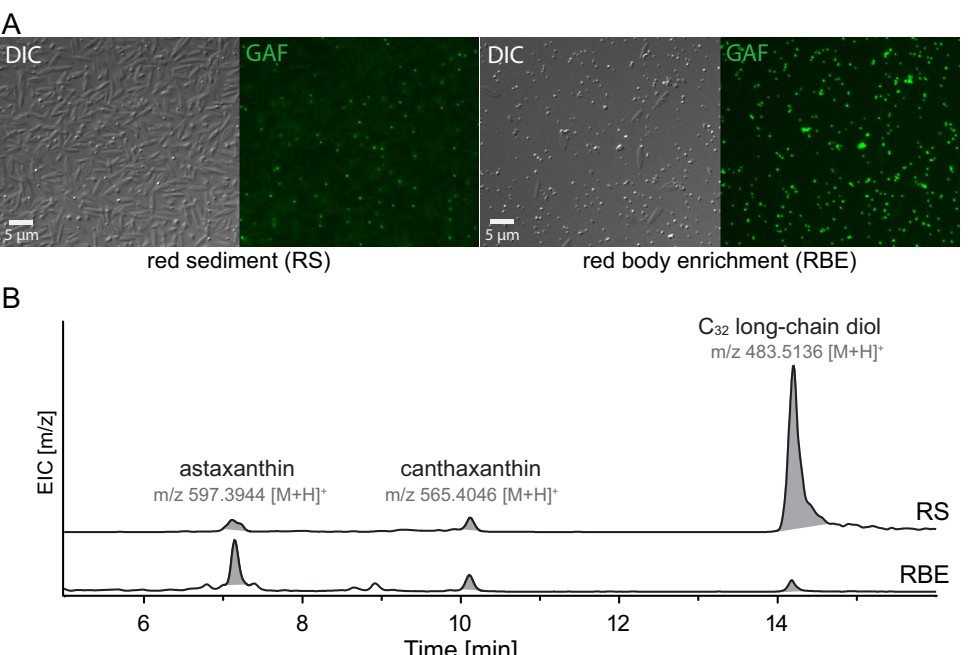

**Fig. 6 | Shed cell walls and red bodies contain ketocarotenoids and a long-chain alkyl diol. A** Light microscopy of the red sediment (RS) mixture and a preparation enriched in the shed red bodies (RBE). Differential interference contrast (DIC) and green autofluorescence (GAF). The scale bar in each pair of images equals 5 μm. Red sediment collection and enrichment were performed numerous times (at least 10) and examined in this way for quality control. Example images from one of those experiments is shown. **B** Extracted ion chromatograms (EIC) from UHPLC-HRMS in APCI mode of organic solvent extracts from red sediment (RS) and red body enrichment (RBE) samples. Shown is the EIC of m/z 597.3944, m/z 565.4046, and m/z 483.5136 ± m/z 0.01 for identification of astaxanthin, canthaxanthin, and $C_{32}$ long-chain diols, respectively. Identity of astaxanthin and canthaxanthin was confirmed by authentic standards. Identification of the $C_{32}$ long-chain diols was based on its accurate mass (<5 ppm) and the observed in-source fragmentation of this compound in APCI (Supplementary Fig. 10).

(UHPLC-HRMS) (Supplementary Data 4 and 5). Consistent with the observed red color of the sample, this analysis revealed the presence of the ketocarotenoids astaxanthin (m/z 597.3944 [M + H]) and canthaxanthin (m/z 565.4046 [M + H]) (Fig. 6B).

Published reports of the chemical composition of *Nannochloropsis* algaenan indicated a structure made primarily of unbranched $C_{30-32}$ alkyl diols (terminal and mid-chain –OH) in which the hydroxyl groups have been converted to ether linkages between alkyl chains[48,57,58]. In *Nannochloropsis*, the precursors to algaenan are thought to be long-chain diols (LCDs), as well as mono- and di-hydroxylated fatty acids that are reduced to form the LCDs[59,60]. In the UHPLC-HRMS, we identified a molecule with a parent mass of m/z 483.5117 [M + H], which corresponds to a predicted molecular formula $C_{32}H_{66}O_2$ (mass error = −3.84 ppm) (Fig. 6B), likely being the LCD dotriacontane-1,15-diol. In the mass spectra of the detected LCD, m/z 465.5012 and 447.4896 were also observed. These correspond to putative in-source fragmentation products of the LCD without one or both of the -OH groups (mass error = −3.65 ppm and −6.32 ppm, respectively) (Supplementary Fig. 10). Relative to the observed amount of LCD, ketocarotenoids were more abundant in the red body enrichment sample, while the opposite was observed in the red sediment sample (Fig. 6B). Because the red sediment contains much more shed wall material, this suggests that, for extractable molecules, the LCD originated primarily from the shed wall, and the carotenoids from the red body.

Accumulation of ketocarotenoids has been observed in several green algae including *Chromochloris zofingiensis*, in which beta-carotene ketolase (*BKT*) catalyzes the addition of the characteristic ketone groups[61,62]. A clear homolog of green algal BKT was not found in the *N. oceanica* genome so we ectopically expressed the *C. zofingiensis BKT* (*CzBKT*ox) coding sequence, with and without fusion to an *N. oceanica* chloroplast targeting peptide (cTP) (Supplementary Fig. 11, Supplementary Data 1)[63]. Only transformation of the cTP fusion construct into *N. oceanica* produced any colonies with a noticeable brown tint, which yielded cultures with an obvious brown color (Fig. 7A). Bulk cell suspensions had increased absorbance between 460 nm−580 nm, suggesting increased carotenoid content (Fig. 7B). Comparing the extracted ion chromatograms from UHPLC-HRMS analysis, the *CzBKT*ox cells yielded more canthaxanthin than astaxanthin relative to wild-type cells, and the intermediate between the two, adonirubin, was only clearly detected in *CzBKT*ox cells (Fig. 7C). Strikingly, we observed unusually large red bodies in these cells, as well as the presence of cells with multiple red bodies (Fig. 7D). Because significant ketocarotenoid accumulation only was observed upon expression of cTP-*Cz*BKT, which directed the enzyme to the plastid where endogenous carotenoid biosynthetic enzymes are located, we interpret these observed effects on the red body as evidence of metabolic flux between the chloroplast and red body.

## Proteomics identifies specific candidate proteins found in isolated, shed red bodies

Pigment extraction of shed red bodies resulted in an insoluble precipitate. Organic solvents are a common precipitation agent for high protein matrices, leading us to investigate the protein content of the red body. Isolated red bodies were completely solubilized when treated with an SDS buffer commonly used for protein extraction for SDS-PAGE. The unstained SDS-gel showed an orange hue from the samples, while staining with Coomassie brilliant Blue confirmed the presence of red body proteins, revealing a distinct band at ~29 kDa (Supplementary Fig. 12). Proteomic analysis of the red body protein content, revealed 20 proteins found in the *N. oceanica* genome, with ≥27 spectral counts (Supplementary Table 1). None of the red body proteins showed homology to proteins identified in previous studies in *N. oceanica* and the diatom *Phaeodactylum tricornutum* lipid body isolates[64–67]. 94 spectral counts were assigned to the gene model jgi|Nanoce1779_2|578349, which has been annotated as an Apolipoprotein D/Lipocalin

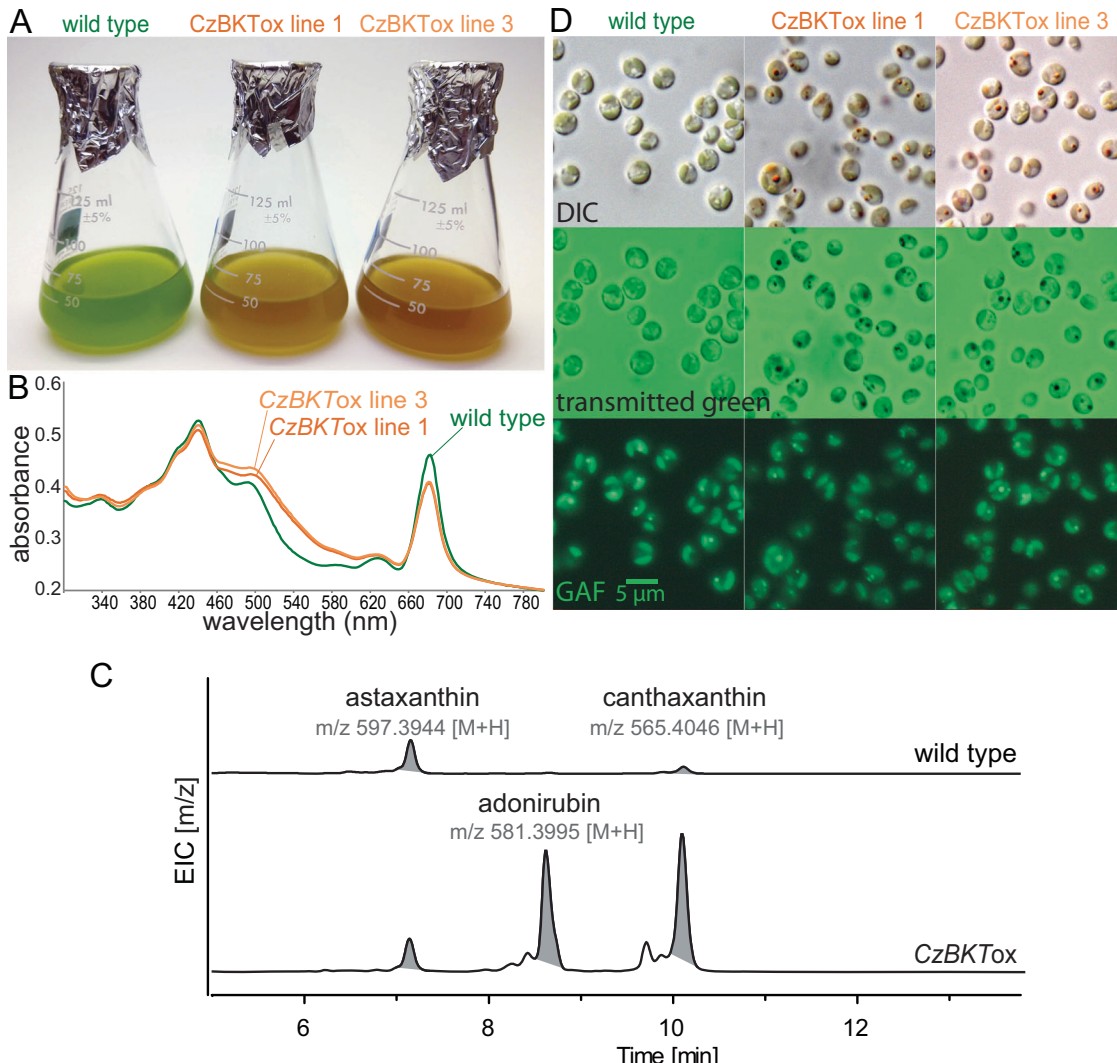

**Fig. 7 | Ketocarotenoid overaccumulation leads to aberrant red bodies. A** *N. oceanica* cultures of wild type and two independent lines overexpressing a beta-carotene ketolase from the green alga, *C. zofingiensis* (*CzBKT*). Cultures were adjusted to equal cell density for imaging. **B** Absorbance spectra for the same cell suspensions, normalized to $OD_{800}$ (primarily scattering). **C** UHPLC-HRMS analysis of extracts from wild type and *CzBKTox* cells (Supplementary Data 2 and 3). Shown is the extracted ion chromatogram (EIC) of m/z 597.3944, m/z 565.4046, and m/z 581.3995 ± m/z 0.01 for identification of astaxanthin, canthaxanthin, and adonixanthin, respectively. Astaxanthin and canthaxanthin were confirmed by authentic standards while adonixanthin identification was based on accurate mass (<5 ppm) and comparison to reference absorbance spectra[133] **D** Light microscopy showing differential interference contrast (DIC), transmitted green light (broad spectrum halogen lamp viewed through the GFP filter set), and green auto-fluorescence (GAF). Scale bar = 5 μm and applies to all images. Cells of these genotypes were collected for at least 3 separate experiments, and representative results were shown.

type protein (NoLCN) (Joint Genome Institute, *N. oceanica* CCMP1779 v2.0 release), with a predicted N-terminal signal peptide targeting NoLCN for secretion (Supplementary Table 1). *NoLCN* was targeted for gene knock-out by CRISPR, and genotyping with gene-specific primers confirmed that three independent *lcn* mutant lines were established (Supplementary Fig. 13). However, no changes in the red body morphology or fluorescence were observed for the *lcn* knock-out mutants. Additional studies are required to resolve the function of the lipocalin in the red body, or to determine whether it is a contaminant from *N. oceanica* cell debris.

**Infrared spectroscopy indicates shed cell walls and red bodies both contain saturated lipids, but differ in carbohydrate and carbonyl-associated molecules**

To obtain even greater enrichment of the shed red body for infrared spectroscopy, we extended the preparation method from the pigment analysis (Supplementary Fig. 9) with ultracentrifugation through a

sucrose density gradient, followed by water washes (Supplementary Fig. 10a and b). Additionally, we found that by incubating the red sediment mixture in 1% SDS at 50 °C, the red bodies were solubilized, leaving behind the shed autosporangial walls (Supplementary Fig. 14). Cultures of another eustigmatophyte, *Goniochloris sculpta*, also contained extracellular debris that appeared to be shed cell walls that could be isolated by differential sedimentation. However, extracellular red bodies from this species were not obvious (Supplementary Fig. 6C). These various isolations were prepared and analyzed by attenuated total reflection infrared (ATR-IR) spectroscopy[68]. Overview spectra with general annotations are shown in Fig. 8 (detailed peak positions in Supplementary Fig. 15; higher order derivative analyzes in Supplementary Fig. 16), and detailed relevant peak assignments are summarized in Supplementary Table 2.

All spectra contained very notable absorbance peaks associated with unbranched, long-chain, saturated lipids. These include a strong pair at ~2916 cm⁻¹ and ~2848 cm⁻¹ (anti-symmetric and symmetric C–H

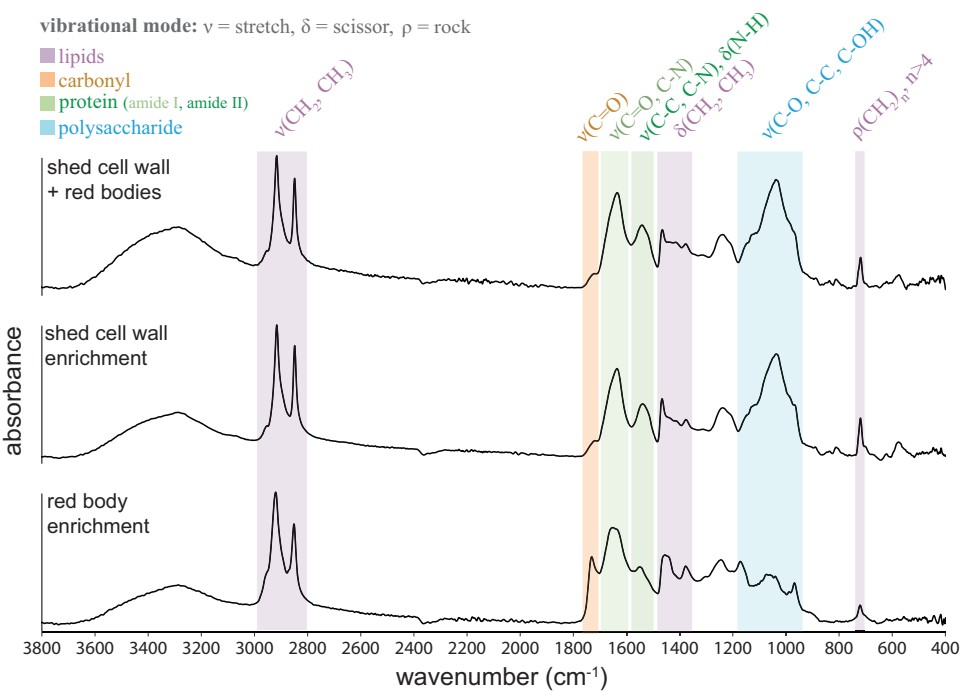

**Fig. 8 | ATR FTIR spectroscopy indicates shed cell walls and red bodies have different lipid and carbohydrate composition.** Attenuated total reflectance Fourier transform infrared (ATR FTIR) spectroscopy of shed cell walls and red bodies (red sediment) and preparations enriched in only the shed cell walls or the red bodies. Baseline corrected absorbance spectra are shown, vertically scaled to equal heights. Molecular vibration assignments are annotated above bands that are colorcoded with their associations to the dominant biological classes of molecules. Classes, their label colors, and approximate wavenumber locations (cm$^{-1}$) are as follows: lipids (purple: 3100–2800, 1500–1300, 1240 and 1171, 721–718); non-peptide carbonyl (orange: 1750–1700), protein (light green for Amide I: 1690–1600, dark green for Amide II: 1600–1480), polysaccharide (blue: numerous features between 1300–900).

stretching modes of methylene groups, $CH_2$) with a small shoulder at ~2954 cm$^{-1}$ (anti-symmetric C–H stretching mode of terminal methyl groups, $CH_3$), a moderate pair near 1460 cm$^{-1}$ and 1375 cm$^{-1}$ ($CH_2$ and $CH_3$ scissoring), and a peak near 720 cm$^{-1}$ (rocking of more than 4 adjacent methylene) (Fig. 8, purple bands)[69,70]. The higher absorption intensity and a red shift of the methylene rocking band from ~721 cm$^{-1}$ in the red body to ~718 cm$^{-1}$ in the shed wall sample indicate a longer alkane chain length in the shed wall.

IR spectra for both red bodies and shed walls showed high-intensity protein bands at 1700–1600 cm$^{-1}$ (C=O stretch in amide I groups) (Fig. 8, light green band) and 1580–1480 cm$^{-1}$ (N–H/C–N stretch in amide II groups) (Fig. 8, dark green bands). The shift of the max of the amide I band from ~1648 cm$^{-1}$ in the red body to ~1635 cm$^{-1}$ in the cell wall implied a change of the dominant protein secondary structure from unordered in the red body to β-sheet in the cell wall[71].

The spectra differed notably in the non-peptide carbonyl region (1750–1700 cm$^{-1}$; C=O stretch) (Fig. 8, orange band). The spectra from red bodies contained a prominent peak at ~1731 cm$^{-1}$ not found in other samples, and further analyzes suggested the presence of long-chain fatty acids, esters/ether esters[72–75]. The peak in the raw spectrum at ~1731 cm$^{-1}$ arose from two overlapping subcomponents at ~1734 cm$^{-1}$ (carbonyl stretch in saturated long-chain esters or ether esters) and ~1710 cm$^{-1}$ (carbonyl stretch in saturated carboxylic acids or ketones) (Supplementary Fig. 16A). The presence of esters and ether ester functional groups was supported by the strong pair at ~1244 cm$^{-1}$ and ~1171 cm$^{-1}$ (C–H deformation of $CH_2$ groups of the long-chain moiety coupled to the C–O vibrations of C–O–C) (Supplementary Fig. 15). For shed walls, the strong carbonyl shoulder at ~1718 cm$^{-1}$ is a summation of two overlapping components with maxima at ~1743 cm$^{-1}$ (carbonyl stretch in acyl groups of saturated triglyceride moiety) and ~1722 cm$^{-1}$ (carbonyl stretch in aliphatic aldehydes or carboxylic acids) (Supplementary Figs. 15 and 16A). As a consistency check, the subcomponent peaks found in only the samples of shed walls, or only red bodies were

all present together in the 2$^{nd}$ derivative spectrum of the red sediment mixture of the two (Supplementary Fig. 16A).

Shed red bodies further differed from shed cell walls in poly-saccharide structure and abundance. Shed walls exhibited a prominent, broad peak centered at ~1038 cm$^{-1}$ (C–O–C stretch of alkyl ether) in the region of 1140–1000 cm$^{-1}$ common to cellulose molecules due to coupling of the C–O/C–C stretch with the C–OH bending or C–O–C ether bridge[76] (Fig. 8, blue band). This typical spectral feature of cellulose was missing in the red body enrichment spectrum. In mechanically isolated cell walls of whole *Microchloropsis gaditana* cells, a similar polysaccharide-associated feature was shown to be greatly diminished by cellulase treatment[48]. These authors also assigned a narrow band at ~1157 cm$^{-1}$ to the glycosidic C–O–C ether stretch, which we also found in higher derivative analyzes of the shed wall enriched sample spectrum (Supplementary Fig. 16C). Altogether, these data strongly suggest the presence of algal cellulose in the shed cell wall and relative scarcity in shed red bodies.

Despite belonging to an entirely different order-level clade within the class Eustigmatophyceae relative to *Nannochloropsis* (Fig. 1)[4], the infrared spectra of shed cell walls of *Goniochloris sculpta* generally resembled those of *N. oceanica*. The peaks from *Nannochloropsis* indicating long-chain, unbranched lipids (e.g. ~2909 cm$^{-1}$ and ~2850 cm$^{-1}$, ~1463 cm$^{-1}$, ~1375 cm$^{-1}$ and ~718 cm$^{-1}$), amide bands (~1634 cm$^{-1}$ and ~1530 cm$^{-1}$), and cellulose (broad peak at ~1043 cm$^{-1}$, narrow peak at ~1159 cm$^{-1}$) were present in the *G. sculpta* cell walls, although with different exact locations (Supplementary Figs. 15 and 16). Of course, in addition to the broad similarities, there were many differences in the exact location and entirely present/absent peaks between the two species (Supplementary Figs. 15 and 16).

The globular form and carotenoid content of the red body suggested similarities between this organelle and pigmented triglyceride storage bodies in other algae[66,77,78]. However, the non-peptide carbonyl peak position from shed red bodies is not consistent with triglycerides

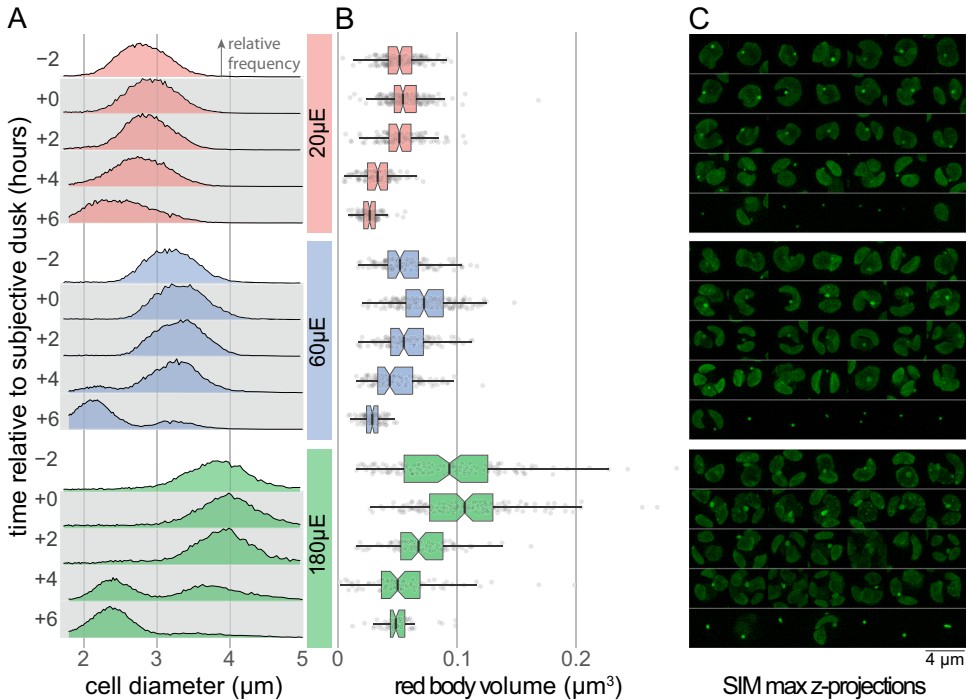

**Fig. 9 | Red body volume is correlated with cell size, and it declines during autospore maturation. A** Synchronous cultures grown under 3% $CO_2$ at light levels of 20, 60, and 180 µmol m$^{-2}$ s$^{-1}$ (abbreviated as "µE"). Cell size was quantified by Coulter counter at the stated time points relative to subjective dusk, and the resulting distributions are shown as relative frequency. Bin size -0.01 µm, maximal y-axis height for whole panel ~3.5%, n (counts) per distribution > 13k. **B** Volumetric estimates derived from SIM z-stacks of cells sampled at the same time points as in (**A**). The notched box plot centers on the median with box hinges representing quartiles. The notch width corresponds to the 95% confidence interval for the location of the median. Upper whiskers extend to largest value no larger than 1.5* the interquartile range. Lower whiskers represent the same, for the smallest value. Individual data points are plotted behind. ANOVA and post-hoc test results for all comparisons included in Source Data. The n=between 56 and 204 for each distribution (exact n included in Source Data). **C** SIM maximum intensity z-projections of randomly selected example cells. 4-micron scale bar applies to all images in (**C**). All panels in this figure share the same time-related y-axes.

---

(Supplementary Fig. 16). Further, we isolated shed red bodies and subjected them to ultracentrifugation though discontinuous sucrose gradients to roughly estimate their buoyant density. These isolated red bodies passed through 24% but not 30% sucrose solution (weight/volume at 20 °C), bounding their density between 1.103−1.127 g/cm³ (Supplementary Fig. 17)[79]. This is considerably denser than would be expected for typical lipid bodies, which sediment on top of aqueous centrifugation layers (0.998 g/cm³)[80].

### Hypothesis: The red body is an organelle that carries algaenan precursors to the apoplast for cell wall biosynthesis during cell division

Our combined observations led us to the hypothesis that the red body is an organelle that carries material into the apoplast during the late stages of cell division and autospore development. These molecules may include plastid-derived lipid precursors of the algaenan polymer and proteins required for its biochemical maturation. The physical and metabolic connections with the plastid, the secretion into the apoplast within the enclosing autosporangial wall, and the presence of long-chain, saturated lipids (matching the expected composition of algaenan precursors) are consistent with this proposed function.

### Red body organelle volume is correlated with cell size and declines during autospore maturation

If the red body accumulates algaenan precursors, we expected that larger parental cells would have larger red bodies, as the resulting autospores would be larger, and require a greater amount of new cell wall material to match the increase in cell surface area. Secondly, we expected that the red body would decrease in volume as material is released during the deposition of autospore cell walls.

By growing cells at elevated $CO_2$ (3%) under three different light intensities (20, 60, and 180 µmol photons m$^{-2}$ s$^{-1}$), we were able to obtain cells of different maximum size (Fig. 9A). Maximum red body volume, as estimated from super-resolution fluorescence microscopy, occurred around subjective dusk, and appeared to increase with cell size (Fig. 9B, C). Cell division was completed by dusk +4 h (using chloroplast division as a proxy, Fig. 9C), and this was distinct from autospore release occurring by +6 h (Fig. 9A). Red body volume decreased after subjective dusk in cells grown at any light intensity (Fig. 9b). Interestingly, at the highest light level, eight autospores per initial cell were frequently observed, compared to the four typically seen in our standard 60 µmol photons m$^{-2}$ s$^{-1}$ condition (Fig. 9C). The central tendency can be informally assessed graphically by the 95% confidence intervals for the median, given by the notches in the box plots (Fig. 9B). The results of an ANOVA and post-hoc comparison test are included in Fig. 9, source data 1, and the compiled measurements themselves in Fig. 9, source data 2.

## Discussion
### The algaenan biogenesis hypothesis for red body form and function

We propose that the red body of *N. oceanica* functions in algaenan biosynthesis by compartmentalizing precursor molecules along with ancillary proteins and small molecules as they are synthesized during the daily growth period. Later, during cell division at night, the red body then delivers these molecules to the apoplast where they generate the outer algaenan layer of autospore cell walls (Fig. 10).

Our chemical analyses support the presence of algaenan precursors in the shed red bodies and provide intriguing connections to previous work on algaenan. *Nannochloropsis* algaenan is thought to be

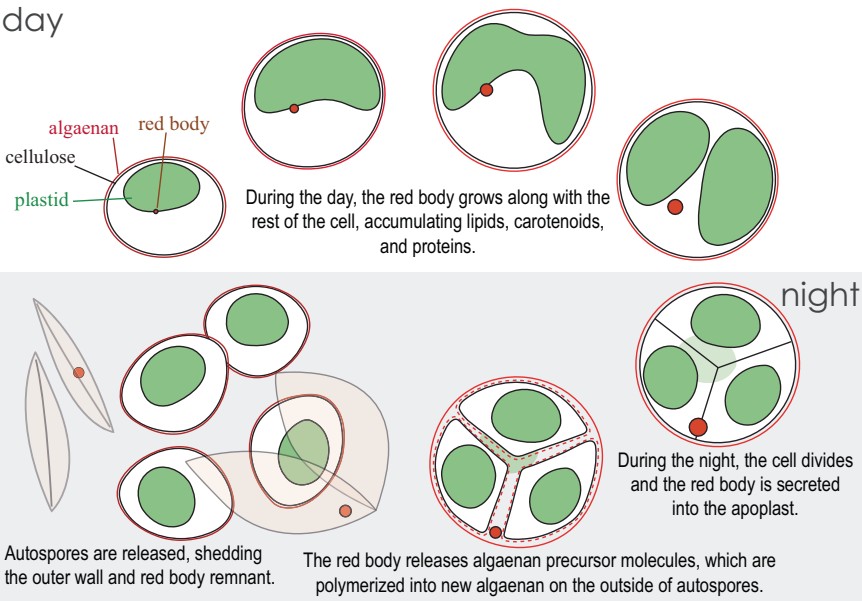

**Fig. 10 | A proposed model for the development and function of the red body in *N. oceanica*.**

composed of long (mostly $C_{32}$), straight, saturated alkyl chains connected by ether cross links, and the immediate precursors are proposed to be LCDs (e.g. $C_{32}$ 1,15-diol)[48,58,81]. These studies inferred the identity of the precursors from the structure of the polymer, and here we directly detected the pseudo molecular ion of $C_{32}$ 1,15-diol and m/z of fragments thereof in organic solvent extraction of shed autosporangial walls, suggesting that there exists some free or loosely associated LCD in these shed walls (Fig. 6). Notably, these m/z were only detected at low levels in the shed red bodies themselves.

However, our infrared spectroscopic experiments showed that the shed red bodies contained methylene-rich signs expected from long, straight alkyl chains, and also a clear enrichment for non-peptide carbonyl groups (Fig. 8). These carbonyls likely belong to saturated esters, carboxylic acids, or ketones (Supplementary Fig. 16), which could represent algaenan intermediates or precursors containing more reactive functional groups. In fact, recent work tracing the biosynthetic pathway to algaenan has implicated long-chain hydroxyl fatty acids (LCHAs) as possible precursors to LCDs, and that the LCHAs themselves are derived from the condensation or elongation of saturated fatty acids[59,60]. These authors also identified a potential wax ester synthase that may catalyze the formation of esters from alcohols and fatty acids[59]. Thus, the infrared spectra of shed red bodies are consistent with the presence of precursors to the LCDs themselves or with intermediate products between LCDs and fully polymerized algaenan.

The localization of the red body is consistent with a role in gathering and dispersing algaenan precursors. The red body is membrane-enclosed (Fig. 2, Supplementary Movie 1) and forms in close association with the chloroplast (Fig. 3), likely enveloped by the chloroplastic ER (Fig. 2). This position would facilitate the influx of the plastid-derived carotenoids that we detected (Figs. 6, 7)[82], as well as the long-chain alcohols, diols or fatty acid precursors that likely also originate in the plastid[59,60]. Formation in the chloroplastic ER would also facilitate lipid modifications to the precursors[83] and incorporation of proteins targeted to the secretory pathway[63]. Indeed, our infrared spectroscopy revealed the presence of proteins in the shed red bodies (Fig. 8), and many of the proteins found in shed red bodies by proteomic mass spectrometry are predicted to have signal peptides for targeting to the secretory pathway (Supplementary Table 1). Some of these possibly function in carrying or polymerizing algaenan precursors or in forming a core-embedded

component of the cell wall[48]. After its secretion (Supplementary Fig. 5), the now-apoplastic red body would be positioned to release algaenan precursors and proteins necessary for polymerization.

The temporal patterns of the red body "lifecycle" in *N. oceanica* are consistent with a role in cell wall biogenesis. The red body is generated de novo each day in healthy, growing cells, and it appears to be secreted around the time of cell division, a few hours prior to autospore release from the autosporangial wall (Fig. 3). This is also the time when cells exhibited increased permeability to exogenous dyes (Fig. 4), which supports the existence of a transition period from protection by the autosporangial wall to coverage by maturing autospore algaenan. In fact, a recent report utilized this period of permeability and showed an approximately 5-fold increase in transformation efficiency when synchronized *N. oceanica* cells were transformed during G2/M phase, compared to unsynchronized cells[84]. Additionally, our hypothesis predicts that the red body would decrease in size during autospore maturation as precursor molecules were released. We informally observed this in 24-hour timecourse observations (Fig. 3), and again when we quantified the red body volume with greater resolution (Fig. 9).

An alternative hypothesis for the function of the red body could be that it serves as an eyespot like those commonly found in other algae. For example, in *Chlamydomonas reinhardtii*, carotenoid-rich bodies serve to directionally shield photoreceptors in order to orient phototaxis[85]. However, this does not explain the red body's function in non-motile *Nannochloropsis* cells or why it would be secreted upon cell division. Secretion of cellular waste has not been extensively examined in *Nannochloropsis*, though media recycling studies indicate the presence of secreted inhibitory molecules[56]. The red body could function in secretion of waste products, however, the chemical similarities between the shed red body remnant and algaenan (Fig. 8) and the observation that the red body decreases in size while still associated with the cell (Fig. 9) are not obviously explained by this waste removal hypothesis.

### Expanding our understanding of the red body and algaenan biosynthesis

Numerous questions still surround the details of the red body biogenesis and function. What drives red body nucleation, and how is cargo targeted to the growing red body? Liquid-liquid phase

separation is recognized as an important driver in organizing biological condensates[86], and extending this biophysical perspective from studies of other types of lipid droplets[87] to the red body may provide clues. The functional advantage of secreting a single, relatively large organelle instead of smaller exosomes is not clear. In fact, inheritance of the red body has been observed for some eustigmatophytes[8], so this may not be a ubiquitous strategy. While the presence of solvent-extractable LCD from shed autosporangial walls suggests that algaenan precursors may be in excess, how the cell wall expands to accommodate growth while the red body is intracellular during the day period and why a remnant red body remains after autospore release are open questions.

Much remains to be determined regarding the composition of the red body and algaenan. The infrared spectroscopy of shed red bodies presented here is consistent with the presence of algaenan precursors or intermediates. Isolating and characterizing specific molecules from shed red bodies, and red bodies isolated from intact cells could take this a step further, and reveal the series of precursors leading to algaenan. Similarly, characterizing the proteins we identified in the red body could provide clues to the enzymes required for algaenan biosynthesis.

Regarding the carotenoid content of the red body, these lipophilic antioxidants might serve to prevent the formation of reactive oxygen species during cell wall biogenesis. Compared to e.g. β-carotene, ketocarotenoids have a higher antioxidant activity in particular in non-liposomal conditions[88], properties that could guide future studies on red body specificity towards ketocarotenoids. Alternatively, the ketocarotenoids could be integrated into the cell wall as structural components. Carotenoids have been found in the cell coverings of several organisms[89,90], including the rapidly growing green alga *Picochlorum celeri*, which produces an extracellular polysaccharide matrix that sediments as a red layer on top of the cell pellet[91] in a strikingly similar manner as the shed autosporangial walls and red bodies described here. Carotenoids are generally presumed to be non-fluorescent[92], though examples to the contrary can be found[90,93–95], and so determining the biophysical mechanism underlying the apparent autofluorescence of the red body may provide insight into carotenoid fluorescence generally.

## Implications for research and applications related to *Nannochloropsis*

The existence of the eustigmatophyte red body may not be known to many researchers, which may influence the interpretation of results, particularly with regards to microscopy. As examples, a hyperspectral imaging study describing carotenoid-rich lipid bodies[94] and a study using the fluorescent dye BODIPY to visualize lipid droplet accumulation[96] in *Nannochloropsis* may have misidentified the autofluorescent red bodies as triglyceride droplets.

The low staining efficiency of cells enclosed by an algaenan cell wall has been noted by other authors[49–51], and fixation of cells is often required to allow staining[97]. Here, we identified a window of greater permeability preceding autospore release that could prove to be useful for technical reasons. This knowledge also might lead to re-interpretation of staining experiments, as the rare cells that stain might be structurally compromised or dead.

According to recent phylogenetic analyzes[4], Fig. 1 represents species from the major clades within Eustigmatophyceae. The occurrence of the red body in these different taxa from many different environments implies that the red body is an evolutionarily ancient characteristic, and the similarities in infrared spectra of *Nannochloropsis* and *Goniochloris* shed walls (Fig. 7) indicate that algaenan may also be a shared, derived feature. Further work with the lesser studied eustigmatophytes will illuminate how this common organelle has diversified in form and function across the clade.

## Eustigmatophytes as model organisms to study the biosynthesis of chemically recalcitrant lipidic biopolymers

Hydrophobic, lipidic biopolymers like cutin, cutan, sporopollenin, and suberin play key roles in plant physiology by regulating the movement of water within and around these organisms. The final chemical composition varies between these polymers, however they share the same upstream, plastid-derived lipid precursors[98,99]. This means that the mechanisms of their biosynthesis also share a common "problem"—how to transport hydrophobic precursor molecules through the aqueous compartments of the cell to the site of polymerization in the extracellular space. Several ABC transporters and lipid transport proteins (LTPs) have been implicated in hydrophobic polymer biosynthesis, but the details of intracellular trafficking and the final polymerization largely remain outstanding[100,101].

The proposed plastid-to-apoplast transport of the red body represents one possible mechanism whereby the canonical secretory pathway may have been adapted to transport a single, large payload of cell wall components to the apoplast. The red body of *Nannochloropsis* may constitute a conceptual parallel with the orbicules of grasses, which are sporopollenin-containing granules found adjacent to maturing microspores in the anther, although the exact function of these bodies is unclear[102]. Similarly, cutin monomers can coalesce into nanoparticles called cutinsomes, which may play a role in bulk transport of cutin precursors through the aqueous environment of the cell wall to the cuticle[103]. Given our characterization of the red body, and the various advantages of *Nannochloropsis* and other eustigmatophyte algae as model organisms (rapid growth, unicellularity, genetic tractability), we propose that they may serve as promising experimental systems for future studies of lipid biopolymers and extracellular metabolism.

Lastly, better understanding hydrophobic biopolymers like algaenan at the biochemical and cellular level may improve our understanding of how they factor into biogeochemical processes at the global level. Such polymers can contribute to long-lived carbon pools, possibly due to their chemical recalcitrance to degradation[104–107]. Increasing inputs of durable plant carbon (e.g. suberin) into soils is being explored as a climate change mitigation approach[108]. Additionally, LCDs likely originating from eustigmatophyte algae persist in freshwater sediments, and they have been studied as useful ecological and geological proxies (e.g. for temperature)[109–111]. We propose that the red body is the physical context connecting LCDs (and their precursors) with algaenan cell walls, and thus it constitutes a new cell biological link in the chain between molecular and large-scale processes.

## Methods
### Statistics and reproducibility
In order to characterize red body shape, cell localization, and autofluorescence across the eustigmatophyte lineage described in Fig. 1, cells of each species were prepared and imaged on at least 3 different occasions with similar results. More detailed imaging of the red body of *Nannochloropsis* through resin-embedded TEM, cells were collected, processed, and imaged in at least 5 separate experiments, with example cells exhibiting good contrast and clarity shown. A single sample was used for the laborious CLEM/cryoET pipeline, with 3 cells examined and one shown in Fig. 2. Experiments determining the red body is shed during cell division, as shown in Supplementary Fig. 5, were performed at least 5 times with slight variations in procedure yielding similar results. Representative example images for each of the separate experiments shown in Fig. 5. Red sediment and fluorescence microscopy was observed numerous times (>20), whole mount negative staining was carried out at least 2 times, and resin-embedded thin sections were prepared at least 5 different times. Red sediment collection and enrichment were performed numerous times (>10) and

examined via microscopy for quality control. Example images from one of those experiments is shown in Fig. 6. Characterization of *CzBKT* genotypes were collected for at least 3 separate experiments, and representative results shown.

## Strains and culture conditions

*Nannochloropsis oceanica* CCMP1779 was obtained from the Provasoli-Guillard National Center for Culture of Marine Phytoplankton (https://ncma.bigelow.org). For all experiments, media consisted of artificial seawater (final solute = 21 g L$^{-1}$) with nutrient enrichment based on full-strength f medium (final concentrations of macronutrients: $NaH_2PO_4 = 83$ μM, $NH_4Cl = 2$ mM) micronutrients as described in ref. 112. Media was buffered by 10 mM Tris-HCl pH 8.1. Liquid cultures were shaken at -100 rpm in sterilized borosilicate Erlenmeyer flasks with perforated aluminum foil caps with a filter paper insert (Whatman #1, United Kingdom) for gas exchange. Small liquid cultures were maintained in polystyrene tissue culture plates (Genesee Scientific, USA). For strain maintenance, cell patches were streaked onto solid media with 0.9% bactoagar in polystyrene petri dishes.

The following strains were ordered from the SAG Culture Collection of Algae (Gottingen University, Germany): *Monodopsis unipapilla* (SAG 8.83), *Vischeria vischeri* (SAG 860-1, formerly *Eustigmatos vischeri*), *Vicheria* sp. (SAG 48.84, formerly *Chloridella neglecta*), and *Goniochloris sculpta* (SAG 29.96). It was found that Bristol media final concentration (mM) [NaNO$_3$ (2.94), CaCl$_2$*2H$_2$O (0.17) MgSO$_4$*7H$_2$O (0.3) K$_2$HPO$_4$ (0.43) KH$_2$PO$_4$ (1.29) NaCl (0.43)] at a pH of 8.0 worked adequately for these strains. *Goniochloris sculpta* only grew as streaks on this media with 0.8% agar. This medium was supplemented with the same trace minerals and vitamins as in f medium for simplicity.

Growth chambers (Percival E41L2, Perry, IA, USA) were set to 28 °C with light from fluorescent lamps (Philips Alto II F25T8) at 60 μmol photons m$^{-2}$ s$^{-1}$ unless otherwise stated. Generally, experiments were carried out with cells grown under ambient air (-0.04% CO$_2$), but for experiments using elevated CO$_2$ conditions, 3% CO$_2$ was achieved with a gas mixer and compressed CO$_2$ cylinder (Alliance Gas Products, Berkeley, CA, USA). For timecourse experiments, two compartments of an algal growth chamber were each set to 12 h light/12 h dark, but with offsets to accommodate reasonable sampling times (e.g. compartment 1 day period = 0:00 to 12:00, compartment 2 day period = 12:00 to 0:00).

## Cell suspension density and sizing determination

Cultures were maintained in an active growth phase (1 x 10$^6$ to 1 x 10$^7$ cells mL$^{-1}$) in preparation for most experiments. Cell density and sizing were measured with a Coulter counter (Beckman Coulter Multisizer 3, CA, USA). Raw cell diameter event counts were converted to relative frequency and visualized as ridgeline plots with the R tidyverse packages (https://tidyverse.tidyverse.org/) and ggridges (https://cran.r-project.org/web/packages/ggridges/index.html). For routine culture maintenance, >1000 counts were acquired to determine cell suspension density; for cell sizing experiments >10,000 were acquired to obtain smooth distributions.

## Light microscopy

Prior to imaging, algal cells adhered to microscope cover glasses, which improved efficiency and image quality by ensuring all cells in the field of view were in the same focal plane and immobile. To adhere the cells, cover glasses (#1.5 thickness, Fisher Scientific, USA) were first etched in 1 molar HCl at 50 °C for at least 4 h, washed with copious amounts of water and dried. Prior to an imaging experiment, etched cover glasses were then placed on a hydrophobic thermoplastic sheet (Parafilm, Bemis Company Inc., WI, USA) and the top side was coated with 50 mg/mL poly-D-lysine (P6407-5MG, Sigma-Aldrich, MO, USA) in 1x phosphate buffered saline for at least 1 hour at room temperature with gentle rocking. Finally, these cover glasses were rinsed in distilled

water and placed in polystyrene microculture plates with algal cell or shed wall/red body suspensions, which were subsequently centrifuged at 500 g for 1–5 min to bind. These cell-coated coverslips were mounted on microscope slides and sealed with clear nail polish, and gently rinsed and dried before imaging.

Widefield fluorescence microscopy was performed with a Zeiss Axio Imager M2, with Zeiss Plan-NeoFluar 100x/1,30 oil objective (Carl Zeiss, Germany). The microscope was fitted with a QIMAGING 01-QIClick-F-M-12 monochromatic 12-bit camera (QI Imaging, CA, USA) for high-sensitivity and Zeiss MicroPublisher 5.0 RTV camera for true-color imaging. Filters used are listed here with the format "name: excitation bandpass midpoint wavelength (nm) / total bandpass width (nm), dichroic, emission bandpass midpoint / total bandpass width". DAPI: 350/50, 400, 425 long pass. GFP: 470/40, 495, 525/50. YFP: 500/20, 515, 535/30. Cy5: 620/60, 660, 700/75. Image processing was done with the ImageJ distribution FIJI[113]. Grayscale fluorescence images were false-coloured and levelled uniformly using batch processing. DAPI images were false-coloured cyan for visibility rather than a true-to-life deep blue.

A Carl Zeiss Elyra PS.1 was used for a super-resolution structured illumination microscope (SR-SIM). Zeiss Zen Blue software controlled acquisition with default optimized z sampling, and was additionally used for SIM processing. The red body autofluorescence was captured with a 488 nm excitation laser line and a 495–550 nm bandpass + 750 nm long pass green filter set. Chlorophyll autofluorescence was captured using a 642 nm laser line and 655 nm long pass filter. Confocal microscopy was carried out using a Carl Zeiss LSM710. Green autofluorescence from the red body was captured using a 488 nm laser line. Red autofluorescence chlorophyll was captured using a 633 nm excitation source.

## 3D reconstruction and volumetric estimation of SR-SIM

SR-SIM z-stacks were analyzed with the 3D rendering and quantitation capabilities of the program Imaris (Oxford Instruments, United Kingdom). A single channel (green autofluorescence) was used for this volumetric analysis, as the red body was distinct from the plastid autofluorescence at the timepoints investigated. Segmentation was semi-automated, with consistent settings used within a light level treatment, but slight adjustments were made between light levels due to differences in size and morphology of the red bodies and were chosen to minimize any obvious disconnected, fibrous/spiky surface artifacts. These Imaris settings included background subtraction (0.1 μm, 4000 arbitrary fluorescence units), seed diameter (0.200–0.350), quality (4000), sphericity (0.5–0.7). Following segmentation, red bodies were manually selected and their statistics exported. These.csv files were compiled and then plotted with the R ggplot2 package [note, the boxplots include a 95% confidence interval around the median with geom_boxplot (notch = TRUE). Base R functions aov() and TukeyHSD() were used for analysis.]

## Resin-embedded transmission electron microscopy (TEM)

Live *Nannochloropsis* cells were either centrifuged at 956 g to form a pellet or were concentrated via filtration. The concentrated cell paste was placed into 2 mm wide by 50 μm or 100 μm deep aluminum freezing hats, then cryo-immobilized using a BAL-TEC HPM-010 high-pressure freezer (BAL-TEC, Inc., Carlsbad, CA). The samples were placed in a freeze-substitution medium made up of 1% osmium tetroxide, 0.1% uranyl acetate, and 5% double-distilled water in acetone. All samples were freeze-substituted following the Quick Freeze Substitution method outlined in[114]. Sample fragments were removed from carriers and infiltrated with Epon resin in successive steps to 100%, using centrifugation and rocking to facilitate exchange[115]. Resin blocks with sample were cured, trimmed and sectioned (70–90 nm) with a Leica UC6 ultramicrotome (Leica Microsystems, Germany); grids were post-stained with 2% aqueous uranyl acetate and Reynold's lead citrate.

Transmission electron microscope imaging was conducted using either a Tecnai 12 120 kV TEM (FEI, Hillsboro, OR) or a JEOL 1200EX 80 kV TEM (JEOL USA, Peabody, MA); images were collected using a Gatan Ultrascan 1000 CCD camera with Digital Micrograph software (Gatan Inc., Pleasanton, CA).

### Plunge freezing

Four µl of *Nannochloropsis* suspensions ($4 \times 10^7$ cells mL$^{-1}$) were pipetted onto a 300 mesh gold Quantifoil grid with 2 µm holes (Quantifoil Micro Tools, GmbH) and manually blotted from the opposite side in a Vitrobot Mark IV (ThermoFisher Scientific, MA, USA) before being plunge frozen in liquid ethane. Grids were snapped into an autoloader assembly (c-ring clipped autogrid) for stability during downstream handling steps.

### Cryo-focused ion beam and scanning electron microscopy (cryo-FIB-SEM)

Plunge frozen grids were imaged via cryo-fluorescence for tracking and correlating the red body structure across EM modalities. To do this frozen grids were loaded into a Linkham CMS196 cryostage (Linkham Scientific Instruments, Tadworth, England) held at -150 °C and placed onto a Zeiss LSM 880 equipped with Airyscan for fluorescence microscopy. Fluorescence and brightfield images were collected at multiple regions across two grids so that targeted cryo-lamellae could be made of cells with prominent red bodies. Following cryo-fluorescence, and detection of autofluorescence in chloroplasts and the red body, the grids were transferred under liquid nitrogen to a Leica Ace 900 (Leica Microsystems, Germany) and coated with 5 nm of platinum prior to being transferred to the Zeiss Crossbeam 540 (Zeiss, Germany). The Zeiss Crossbeam 540 was equipped with a Leica CryoStage cooled to −150 °C, used for milling and imaging of the frozen cells. Zeiss Atlas and Zen Connect software was used to navigate and map locations to be imaged for correlative cryo-fluorescence-electron microscopy. To create cryo-lamellae for cryo-electron tomography, milling was performed using a gallium ion source at an energy of 37 pA and a working distance of 5 mm. Imaging of the grid and milled lamellae was done using an Everhart-Thornley secondary electron detector at 2.0 kV. When slicing and viewing during the milling run, to generate a three-dimensional image stack (cryo-volume EM), the z-depth for each mill slice was 20 nm, and the target lamella thickness was 400 nm.

### Cryo-electron tomography (cryo-ET)

FIB-milled grids, each with 2 to 4 sites of cryo-lamellae, were transferred to a Titan Krios electron microscope (ThermoFisher Scientific) operated at 300 kV and equipped with a BIO Quantum energy filter and a K3 direct electron detector (Gatan). Full -60 to +60 bi-directional tilt series were collected for 3 lamellae in 2-degree increments with a total dose of 100 e-/Å2 in low dose mode in SerialEM[116]. A total of 5 tilt series were aligned in IMOD[117] using patch tracking and tomograms were reconstructed using the simultaneous iterative reconstruction technique (SIRT) algorithm. After inspection, the best tomogram was selected for its quality. Subsequently, 2D sections as well as videos of the 3D tilt series reconstructions or tomograms were generated in FIJI. The entire 3D cryo-tomogram as well as tilt series data are available in the Electron Microscopy Data Bank (EMDB) archive under the accession code EMD-43893. The 350 nm thick 3D tomogram was then segmented using the TomoSegMemTV membrane detection software package[118]. The red body envelope membrane could be confirmed; its 3D model, as well as the neighbouring cellular compartments is shown in Supplementary Fig. 4.

### Fluorescent labelling

Calcofluor white (CFW, Fluorescent Brightener 28/Calcofluor White M2R; Sigma F-3397 FW 960.9, dye content 90%) binds to β-glucans,

and was used to fluorescently label the cellulose cell wall component (Bidhendi et al. 2020). A 500x stock was made with water, and diluted in *f* medium to the indicated final concentrations (% weight/volume). The DNA-binding dye Hoechst 33342 (AdipoGen CDX-B0030-M025, CA, USA) was used to label nuclei at a final concentration of 5 µg/mL. For the time-dependent permeability experiment, the dye stocks were introduced to the cell culture in 50 mL conical centrifuge tubes. At each time point, and for each treatment, 1.6 mL of this stained suspension was transferred to a microfuge tube and centrifuged at 2000 g x 30 s. Cells generally adhered to the side of the tube and did not form an obvious pellet, but pipetting off 1 mL and repeating the centrifugation led to a small loose pellet. Higher centrifugation forces and times appeared to result in higher background fluorescence and abnormal cell morphology (data not shown). The remaining supernatant was removed and replaced with 0.8 mL fresh *f* medium. This resuspension was centrifuged for 2000 g x 30 s, the supernatant removed, and the pellet resuspended in 1.6 mL fresh *f* medium.

### Bulk fluorescence quantification

Fluorescence measurements were collected with an Infinite M1000 Pro plate-reading spectrophotometer/fluorometer (Tecan, Switzerland). Measurements were collected in 96-well, flat optical bottom, black-walled plates (Thermo Fischer Scientific 165305). From the stained and washed cell suspensions, 200 µL was pipetted into each of 7 wells as technical replicates, and fluorescence measured with the following settings: Bottom read, Mode 1 (400 Hz), 50 flashes. Chlorophyll fluorescence- ex/em 630/680 bandwidth=l 10 nm each. Gain = 100. Calcofluor white- 380/475, 10 nm/20 nm bandwidth. Gain = 50. Hoechst- 361/497. 10 nm/10 nm bandwidth. Gain = 100. Absorbance-750 nm. 25 flashes, 100 ms settle time. The experiment was repeated in its entirety three times. Data was compiled, plotted, and analyzed in R; simple linear regression for plotting and analysis of covariance (ANCOVA) for testing differences in slope were carried out using the averaged technical replicate values.

### Sucrose density gradients and ultracentrifugation

Discontinuous sucrose gradients for further purifying shed red body particles were created by manually underlaying denser solutions of sucrose in water beneath less dense ones with thin-tipped glass Pasteur pipettes. Pre-enriched preparations of shed red bodies suspended in water were applied to the top of the gradient and centrifuged for 30 min at 40,000 g in an Optima XE-90 Ultracentrifuge (Beckman Coulter, USA) with SW60 Ti rotor and Beckman Ultra-Clear 11 x 60 mm centrifuge tubes (#344062).

### Generation of *CzBKTox* carotenoid mutants

*Chromochloris zofingiensis* cDNA was isolated as described in ref. 61, and kindly provided by Daniel Westcott. From *C. zofingiensis* cDNA, *CzBKT1* (AY772713)[119] was isolated with gene-specific primers in which a 3xFLAG-tag was introduced at the 3' end. *CzBKT1*, with and without *N. oceanica*-specific chloroplast targeting sequence (cTP)[63], was assembled with a hygromycin resistance cassette (HygR)[23], *CAH1* promoter and *ARF* terminator[28] in pDONR221, by Gibson assembly (Invitrogen, USA). HygR-*CzBKT* constructs with and without cTP were linearized by PCR prior to transformation into *N. oceanica*[23].

### Ultra High performance liquid chromatography with diode array detection & high-resolution mass spectroscopy (UHPLC-DAD-HRMS)

Pigment extraction was done with acetone on lyophilized samples. Acetone was removed by evaporation with N$_2$ and pigments were resuspended in 90:10 methanol:water with 2 ppm 8-apo-carotenal as internal standard. 5 min sonication in water bath (BRANSONIC 5510E-MT) improved pigment resuspension. All samples were filtered (Filter PLT 96, 400 µL PVDF 0.2 µm LNG, Agilent Technologies)

prior to analysis UHPLC-DAD-HRMS analysis was done using the method described in[120]. In brief, the sample extract was analyzed using an Dionex UltiMate 3000 Quaternary Rapid Separation UHPLC + focussed system (ThermoFisher Scientific) coupled to a QTOF Compact mass spectrometer (Bruker, Bremen, Germany). Samples were separated on a ACQUITY UPLC HSS C18 SB column (100 × 2.1 mm ID, 1.8 μm particle size, 100 Å pore size; WATERS; Milford, MA) maintained at 40 °C with a flow rate of 0.5 mL min-1 and mobile phase consisting of A: 50:22.5:22.5:5 water + 5 mM ammonium acetate:methanol:acetonitrile:ethyl acetate, B: 50:50 acetonitrile:ethyl acetate. For detection diode array detection (DAD) UV-Vis 190–800 nm and Atmospheric pressure chemical ionization (APCI) was used as ionization technique for the mass spectroscopy (MS) detection. Briefly, the corona discharge current was set at 5 μA, the vaporizer and capillary temperature were set at 450 and 350 °C, respectively and a capillary voltage of ±25 V was used for both positive and negative APCI. A scanning range from m/z 100 to m/z 1400 was used for the MS in all analytical methods. Data was visualized using DataAnalysis Version 4.3 (Bruker Compass DataAnalysis 4.3 (x64), Bruker Daltonik GmbH). Total number of samples analyzed was 4 (n = 1). Identification of canthaxanthin and astaxanthin was confirmed by authentic standard acquired at DHI group A/S (Hoersholm, Denmark). Identification of adonirubin was supported by comparison to reference absorbance spectra and accurate mass [<5 ppm][121].

## Mass spectrometry proteomics of shed red bodies

Red bodies shed during cell division were collected by centrifugation and isolated from shed walls by high-pressure homogenization and differential centrifugation. The red body enrichment was then solubilized and denatured by heating to 95 °C for 60 s in a protein extraction buffer (40 mM Tris-HCl pH 6.8, 12.5% glycerol, 1% SDS, 0.005% bromophenol blue). This extract was subsequently submitted to polyacrylamide gel electrophoresis (SDS-PAGE) (Mini-PROTEAN TGX Any kD- BIO-RAD) followed by in-gel trypsin digestion. Briefly, the gel slices were subjected to various washes containing $NH_4HCO_3$, DTT, iodacetamide, acetonitrile and dried in a speed vac. Gel pieces were rehydrated and incubated with 0.2 μg of trypsin (Promega, sequencing grade) in 20 μL of buffer overnight at 37 °C. Protein mass spectrometry was carried out at the University of California Berkeley's qb3 Vincent J. Coates Proteomics/Mass Spectrometry Laboratory using a Thermo LTQ XL linear ion trap mass spectrometer. Mass spectrometry was performed at the Proteomics/Mass Spectrometry Laboratory at University of California, Berkeley. Multidimensional protein identification technique (MudPIT) was performed as described[122,123]. Briefly, a 2D nano LC column was packed in a 100-μm inner diameter glass capillary with an integrated pulled emitter tip. The column consisted of 10 cm of Polaris c18 5-μm packing material (Varian) and 4 cm strong cation exchange resin (Partisphere, Hi Chrom). The column was loaded and conditioned using a pressure bomb. The column was then coupled to an electrospray ionization source mounted on a Thermo-Fisher LTQ XL linear ion trap mass spectrometer. An Agilent 1200 HPLC equipped with a split line so as to deliver a flow rate of 1 μl/min was used for chromatography. Peptides were eluted using a 4-step gradient with 4 buffers. Buffer (A) 5% acetonitrile, 0.02% heptafluorobutyric acid (HFBA), buffer (B) 80% acetonitrile, 0.02% HFBA, buffer (C) 250 mM NH4AcOH, 0.02% HFBA, (D) 500 mM NH4AcOH, 0.02% HFBA. Step 1: 0–80% (B) in 70 min, step 2: 0–50% (C) in 5 min and 0–45% (B) in 100 min, step 3: 0–100% (C) in 5 min and 0–45% (B) in 100 min, step 4 0–100% (D) in 5 min and 0–45% (B) in 160 min. Collision-induced dissociation (CID) spectra were collected for each m/z. Raw data can be accessed on www.proteomexchange.org, ID: PXD052104. Protein identification, quantification and analysis were done with Integrated Proteomics Pipeline-IP2 (Bruker Scientific LLC, Billerica, MA, http://

www.bruker.com) using ProLuCID/Sequest[124,125], DTASelect2[126,127], and Census[128,129]. Spectrum raw files were extracted into ms1 and ms2 files from raw files using RawExtract 1.9.9 (http://fields.scripps.edu/downloads.php) 10, and the tandem mass spectra were searched against human database[127,130]. The identified peptide fragments were mapped to the N. oceanica CCMP1779 version 2.0 predicted proteome (available from https://phycocosm.jgi.doe.gov/Nanoce1779_2/Nanoce1779_2.home.html), using DTASelect v2.1.4[126]. The gene models were subsequently ranked in order from the highest number of summed observed spectral counts across the entire protein. Predicted subcellular targeting was done with the full length predicted proteins using HECTAR[131], a heterokont/stramenopile-specific localization tool. The euKaryotic Orthologous Groups of proteins (KOG) database was used to provide potential gene functional annotations (mycocosm.jgi.doe.gov/help/kogbrowser.jsf).

## Generation of CRISPR/Cas9 knock-out mutants of putative lipocalin

A protein corresponding to protein 578349 (N. oceanica CCMP1779, version 2 assembly) was targeted for CRISPR/Cas9 disruption using a pair of gRNAs expressed from an episomal Cas9 vector[25]. 1st gRNA: 5'- TGGAGACTTTGCCCGAGATG -3', 2nd gRNA: 5'- TTCAGTGCGG-GAGATTTTGA -3'. Transformant colonies were selected on hygromycin (50 micrograms/mL) and genotyped by PCR to identify strains with aberrant amplicon sizes, which would indicate mutations at the lipocalin locus. Genotyping primers were F: 5'- AAAGCAAACCGTG-GAAGATG -3', and R: 5'- GCCCGCTTAGACATTCACAT -3', which is expected to generate a 1458 base pair amplicon from wild-type genomic DNA. The 1st gRNA site is located in the first coding sequence (CDS) exon, and the 2nd gRNA site is located in the 3' untranslated region (UTR). The distance between the first CDS and 3' UTR gRNAs is 1061 base pairs, so a full deletion of this size between the gRNAs would produce an approximately ~400 base pair amplicon. Mutant strains were examined by widefield fluorescence microscopy for the red body autofluorescence, as previously described in this manuscript.

## Fourier transform infrared spectroscopy (FTIR)

FTIR spectra were collected with a Vertex 80 Time-resolved FTIR (Bruker, MA, USA) in attenuated total reflectance mode (ATR). 100 spectra from 4000 to 400 cm$^{-1}$ were collected for each sample at a resolution of 4 cm$^{-1}$. Adhesive-backed aluminum foil used for sealing 96-well PCR plates was cut into strips and adhered to a glass microscope slide for ease of handling. Aluminum provides an inexpensive substrate with low background for ATR FTIR[68]. These were cleaned with ethanol and stored with desiccant until use. Concentrated "red sediment," red body enrichment ("red sediment" subjected to high-pressure homogenization, sucrose density gradient and water washes- see Supplementary Fig. 14), and red body-depletion suspensions (SDS incubation of "red sediment" followed by water washes) were spotted onto the foil (5 μL) and placed in a vacuum bell with desiccant until the water was evaporated (>30 min). Samples were stored in a microscope slide holder in a sealed container with desiccant at 4 °C until spectra collection later that day. A position on the foil was left empty to serve as a calibration blank prior to other measurements, and two separate sample spots for each sample type were measured with similar results.

All ATR FTIR data were subject to preprocessing that included conversion from reflectance to absorbance and baseline correction. Second derivatives of the absorption spectra were calculated as a slope through seven neighboring points using Savitsky-Golay in the OMNIC software program (version 9.8, Thermo Fisher Scientific). Peak position determination and data interpretation was guided by the second derivative spectra (Supplementary Fig. 15), published literature, and the NIST open-source library[132].

**Reporting summary**

Further information on research design is available in the Nature Portfolio Reporting Summary linked to this article.

## Data availability

The 3D cryo-tomogram as well as tilt series data for the in situ red body electron microscopy have been deposited in the Electron Microscopy Data Bank (EMDB) archive under the accession code EMD-43893. Raw Red Body Proteomics data have been deposited with ProteomeXchange under the accession code PXD052104. All other data generated in this study are provided in the Supplementary Information/Source Data file. Materials requests and correspondences should be addressed to K.K.N. Source data are provided with this paper.

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

## Acknowledgements

We thank Drs. Steven Ruzin and Denise Schichnes for assistance and consultations with the light microscopy presented here. We thank Joel Mancuso, Vimal Gangadharan, and Jeffrey Marshman of Zeiss Inc. and Abigail Lytton-Jean and David Mankus of the KI Nanomaterials Lab at MIT for their time, expertise, and assistance in cryo-fluorescence and cryo-FIB-milling. Valuable feedback during figure preparation was provided by many colleagues, especially Dhruv Patel-Tupper, Simon Álamos, and Lorenzo Washington. We thank Dr. Daniel Westcott for providing *C. zofingiensis* cDNA, Dr. Masakazu Iwai for providing valuable training and advice regarding sucrose density gradients, Dr. Jian-Hua Chen for performing a preliminary soft x-ray tomography experiment, and Dr. Maria Flores for advice on tomogram segmentation. C.W.G. was supported by the National Science Foundation (Graduate Student Research Fellowship Grant DGE 1106400). H.Y.N.H. was supported by the DOE/BER BSISB program. K.K.N. is an investigator of the Howard Hughes Medical Institute. This article is subject to HHMI's Open Access to Publications policy. HHMI lab heads have previously granted a nonexclusive CC BY 4.0 license to the public and a sublicensable license to HHMI in their research articles. Pursuant to those licenses, the author-accepted manuscript of this article can be made freely available under a CC BY 4.0 license immediately upon publication.

## Author contributions

C.W.G., J.A-R, and K.K.N. designed experiments and developed conceptual directions for this work. J.A-R. performed pigment analysis and UHPLC-HRMS with associated analyzes. E.B. offered critical comments and feedback on drafts of the manuscript, and aided in the preparation of the manuscript for publication. D.J. directed C.W.G. in TEM and performed the cryo FIB-SEM and preparation of the lamellae for cryo electron tomography, which was performed and analyzed by P.G. R.Z.R provided technical insight and direct assistance in analytical chemistry methods, including FTIR. H.N.H. performed the FTIR analyzes, and substantially contributed to the writing of the related manuscript sections. C.W.G. performed the remaining experiments, wrote the manuscript, and prepared the figures with support from J.A-R, H.N.H., and K.K.N. All authors reviewed the final version of the manuscript.

## Competing interests

The authors declare no competing interests.
