## [Peer Review File · Nature Communications]

REVIEWER COMMENTS

Reviewer #1 (Remarks to the Author):

This is a very well written manuscript describing a state-of-the-art characterization of the red body in *Nannochloropsis* and related Eustigmatophytes. Most of the analysis is based on sophisticated microscopy and one of the key findings is that the spot changes in sized in parallel to cellcycle/cell division in a diurnal manner. Compositional analysis on isolated and partially purified red bodies is provided using fluorescence, mass spectrometry and FTIR spectroscopy to show that ketocarotenoids and long chain alcohols, possible precursors of the cell wall material algaenan, are present in the red spot. Overexpression of a cDNA encoding an enzyme involved in ketocarotenoid biosynthesis from a different alga and targeted to the plastid resulted in enlarged red bodies. Based on these data, a new hypothesis was postulated that red bodies are involved in carrying algaenan precursors from the plastid to the apoplast where they are incorporated into the cell walls of newly formed daughter cells.

Overall, this novel hypothesis on red body function is very intriguing and plausible based on the data. Most of the data are descriptive or correlation-based, and the hypothesis raises a lot of questions regarding mechanisms. For example, the data show that the red body is surrounded by a membrane but how the red body actually forms inside the plastid envelope system or how the ketocarotenoids leave the plastid remains to be seen. Some obvious follow up experiments would include proteomics of the purified red bodies to identify possible enzymes or proteins involved in red body formation to generate a more mechanistic hypothesis and targets for gene disruption to test this hypothesis. Furthermore, lipidomics of the red body would help to clarify whether neutral lipids or plastid membrane lipids are present that could provide further hints towards the origin of the red body.

Given that I do not see any technical issues with the provided analysis, the methods used, or the presentation of the data, the main question is whether there is enough novelty, general interest, and support for an intriguing new hypothesis on red bodies in Eustigmatophytes to warrant publication. My personal answer to this question is a resounding yes. General interest will probably be based on the fact that *Nannochloropsis* is an merging model system for biotechnological applications, and the intriguing biology and evolutionary history of Eustigmatophytes.

As a minor note, the authors may want to replace plasmid with plastid in line 104.

Reviewer #2 (Remarks to the Author):

I have attached below a pdf that contains my review

Reviewer #3 (Remarks to the Author):

The presence of red bodies in Eustigmatophyceae including *Nannochloropsis* has been observed previously by other researchers. The MS “A proposed function for the red body of *Nannochloropsis* in the formation of the recalcitrant cell wall polymer, algaenan”, taking advantage of the emerging model marine alga *Nannochloropsis oceanica*, performed a biochemical characterization of red bodies and proposed their function in transporting algaenan precursors from plastid to apoplast to be incorporated into daughter cell walls. While providing well-presented biochemical data, the study lacks genetic proofs to support the conclusion.

1. Where does the red body originate from? Based on the carotenoid profiles of red body, it seemed that only ketocarotenoids were present. Why no other carotenoids like beta-carotene? Or why red body only transports ketocarotenoids out of the plastid? Where is the C32 alkyl diol synthesized?
2. The authors collected red bodies for lipid and carotenoid profiling. These red bodies should also be used for proteomic analysis, with an aim to probe the difference compared to lipid bodies and possible proteins involved in red body biogenesis.
3. Red bodies harbor C32 alkyl diol, the possible precursor of algaenan; this can give an indication but is not solid enough. Additional evidences should be included to underpin the conclusion. Are there ways to impair red body biogenesis (based on the proteome) ? Such mutants would be of help for the authors to confirm the role of red body in algaenan formation.

Reviewer #4 (Remarks to the Author):

I was asked to evaluate the quality of the electron microscopy (tomography) data of the manuscript “A proposed function for the red body of *Nannochloropsis* in the formation of the recalcitrant cell

wall polymer, algaenan” (NCOMMS-23-46997). In this context the methods used and data shown in the manuscript are excellent and state-of-the-art. Besides high pressure freezing to prepare resin embedded samples for TEM investigations the authors also used Cryo-FIB-SEM to image plunge frozen samples as close as possible to their natural state. Both methods produced excellent images of the ultrastructure of the red bodies. Specifically Cryo-FIB-SEM is technically very challenging and the images/videos produced by the authors are outstanding. The conclusions (red bodies are surrounded by a membrane, associated with the chloroplast, red bodies are shed in the apoplast, etc.) are well supported by the images and very interesting. Thus, I can conclude that the methods used and the quality of the images/data from the electron microscopical images are excellent and well suited to be published in Nature Communications. I also have extensive experience in confocal laser scanning microscopy. The methods used and the quality of the images of the light microscopical work is also excellent and suitable for publication in Nature Communication. Since I am not a specialist in the field of red bodies and algae other reviewers need to decide if the scientific value of the observations merits publication in Nature Communications. As already mentioned the quality of the methods and data of the electron and light microscopical work is definitely worth being published in Nature Communications. However, there are some major issues with the manuscript and the authors would need to address these before the manuscript can be published in Nature Communications.

Major issues:

The use of tomography throughout the manuscript that I have received is misleading and should be erased when referring to the electron microscopy studies. Tomography usually results in a 3D reconstruction of the investigated structure which is lacking. I was looking for a 3D reconstruction of a tilt series of the red body in the Cryo-TEM but could not find one even though it is described in the material and methods sections – it might have gone lost during the upload. Then I was looking for a 3D reconstruction of the red body (based on serial sections) imaged by Cryo-FIB-SEM but could not find one either. All I found was a video (10 seconds) that showed several sections (probably 50 sections or so produced with Cryo-FIB-SEM) through an algae cell including half (not a whole) of a red body but the 3D reconstruction which is announced in the manuscript is missing. Did the authors perform 3D reconstruction of a whole/parts of a red body? If yes where is it in the manuscript? What do the authors mean by tomography? Tomography is usually a tilt series of one section (up to 300nm in thickness) in the TEM which is then used for 3D reconstruction. This is definitely not provided in the manuscript that I received. Also if the authors want to perform a 3D reconstruction of the red body (based on serial sections) then they would need to slice/mill through a whole one (currently only half of one is sliced through with Cryo-FIB-SEM) and then reconstruct it with Image J or similar software. Please clarify this issue. The data is already impressive as it is and a 3D reconstruction (even though it would be even more impressive) is not necessary as it would not really support the data any further.

Video 1: The uploaded video 1 (10 seconds) does not show a reconstruction of a 400nm thick lamella and definitely not Cryo electron tomography. I am not sure if an incorrect file was uploaded or if the authors don't understand the term tomography. It would be nice to provide a figure based

on Video 1 where the 3D reconstruction that is claimed in the figure legend of Video 1 is provided. Also, it is unclear what methods were used to produce video 1 (I assume the video shows Cryo-FIB-SEM milling). The method should be described in the figure legend of video 1.

Scale Bars, figure labels, image arrangement etc.: Please read the comments below, go through all of your figures, and adjust scale bars, add labels, etc. according to my comments in all images.

In Figure 1 the authors mention at the very end that the scale bar in the first image applies to all other images. It would be nice if the authors could include a similar sentence in all other figures where this applies. For example the legend of Figure 1 – S1 does not even mention the scale bar. In Figure 1- S2 it is unclear if the scale bar in the first image applies to all other images.

Scale bars in Figure 2 and Figure 6 are at different x,y positions (some are in the left corner some are in the right corner of the image) and different width. Please arrange them properly at similar positions (either left or right bottom corner with same distance from the bottom and side) relative to the image. I personally think that the scale bar in the 2nd image of Figure 2 of the second row is too long (almost the width of the whole image). Either way please arrange the scale bars so the figure looks more appealing.

Figure 2: Images in the second row don't have a label. Please use labels (D, E, F) in the images of the second row. Also I find the labeling for organelles (colors, lower case letters) unusual. I personally would use white or black upper case letters and label the mitochondria with "M", the nucleus with "N", the chloroplasts with "C" and the red body with "RB" throughout the manuscript (including the video). That is how it is usually done in electron micrographs. It is also unclear which images derive from resin embedded samples and which one are from Cryo-FIB-SEM. This should be clearly stated in all the figure legends. I also think that the figure legend should be expanded to clearly describe the images in more detail. The authors need to keep in mind that some readers will have very little background in (electron) microscopy and will not know which methods were used to produce the images and what they are looking at. Especially since almost all of the data in this manuscript derive from micrographs, proper figure legends are essential to make this manuscript impactful.

Figure 4 S1: please include scale bars.

Figure 5: size of scale bar is unclear; please specify.

Figure 6: This is a very ugly figure and must not be published in its current form! The figure legend also lacks a basic description of the findings displayed in the images. This figure would get an F (probably 0 points) in my SEM/TEM class. Here are a few suggestions to improve the figure: please arrange scale bars in the different images in similar x,y positions (bottom right or left corner, same distance from the bottom). Please use the same font for the description of the size of the scale bars. Please arrange the description of the scale bars uniformly either on top or below of all scale bars. Please label all images with a letter (right now only a few are labeled with A, B, C, D). The abbreviation DIC is not explained in the figure legend – please explain. Please introduce some space between the images so that it is clear where one image ends and the other one starts. Please crop and arrange the images properly. Also it is unclear what the images show and which microscopical technique was used. Please include this in the image description so that the reader can understand and not guess what is depicted in the images.

Figure 7: Scale bar is too small and should be made larger. It will be impossible to read the size of the scale bar in the final pdf. Please include the scale bar size in the figure legend.

458125 A proposed function for the red body of *Nannochloropsis* in the formation of the recalcitrant cell wall polymer, algaenan

Christopher W. Gee, Johan Andersen-Ranberg, Rachel Z. Rosen, Danielle Jorgens,
Patricia Grob, Hoi-Ying N. Holman, Krishna K. Niyogi

REVIEW (from Ursula Goodenough)

This is an excellent paper, with many creative experimental approaches, in-depth analyses, and outstanding writing and figures. I'm not qualified to assess the biochemical analyses and trust another reviewer is so endowed. It would be useful to learn to what extent these analyses comport with vs. diverge from those reported in Scholz et al for *gacitana*.

I'll first offer a few comments on the paper and then some unpublished data from our lab that might be useful to the authors.

Possibly the authors hit a limit in references cited, but if not, they might consider adding this:

Wolf et al. (2018) *Photosynth. Res.* 135:177-189.

And this should for sure be included as it has IMO standard-EM images of the Nanno wall:

Murakami R, Hashimoto H. 2009. Unusual nuclear division in *Nannochloropsis oculata* (Eustigmatophyceae, Heterokonta) which may ensure faithful transmission of secondary plastids. *Protist* 160:41–49. <http://dx.doi.org/10.1016/j.protis.2008.09.002>

Line 99. Make clear here that this is max diameter observed and that evidence for smaller sizes is given below.

Line 137 should be 4C.

Line 174 IMO it is misleading to say that the maternal autosporean wall undergoes degradation; instead, it splits open and then lasts forever!

Line 178 The red sediment is also reported in Scholz et al.

Line 334 I'm surprised that an SDS gel of the protein fraction isn't provided to give some idea as to whether it's many faint bands or a few prominent ones.

Unpublished data (I'm happy to provide the Niyogi lab with any full-res images they're interested in.)

In Scholz et al we usually washed out the carotenoids, concerned that they might be contaminants. After publication, however, Taylor Weiss ran a carotenoid analysis of shed walls which supports your analysis. I'm not suggesting that his data be included but it's always nice to see confirmation! We note presence of amino acids in pressed walls (Table 4), which are not expected to contain exocytosed red bodies, suggesting they're in wall itself as well as in red body.

Our QFDE images of walls (Fig. 2 of Scholz) shows only walls released by French Press but in fact we also looked at shed mother walls:

These show that the cellulose layer is retained, confirming your Calcoflor data. Our shed walls curl around whereas yours are straight (your Fig.6); difference might be gaditana, different suspending media (all our wall preps are in water), or something else.

To this point, our pressed walls do not curl up (Fig 2 Scholz). My interpretation is that the mother walls are released via a longitudinal slit, akin to these shells, and they then curl back

on themselves. If this is the case, then presumably most of the red bodies they might have harbored would be released, explaining why they are apparently absent from the images.

So how does this slit work? Here are some musings. A proto-slit might form prior to daughter cell formation, where this entity might account for the increase in permeability you observe. Splitting might occur in conjunction with, or subsequent to, red body release, with a portion of its contents containing an autolysin-analogue that opens up the proto-slit. Your cool idea that the red body also releases daughter-cell algaenans in that interval is totally concordant.

Here are several images that may represent the red body poised for secretion or undergoing secretion. The first two are from Santos and Leedale (Nova Hedwigia 60:219, 1995) and Suda et al (ref in your paper)

Figs 7-14. *Nannochloropsis* species. TEM. C = chloroplast; L = lamellate vesicle; M = mitochondrion; N = nucleus; P = pyrenoid-like

The rest are from various Nannos:

W124-03008 is particularly interesting due to its striated contents. In your video you label an organelle as a red body, but next to it is a similar organelle with striated contents:

I have numerous images with striated interiors, most in cells that were N-starved to induce TAG production. Here's one:

So this suggests a story: that when cells enter stationary via N-starve, the algaenan in their red bodies goes ahead and creates layers of outer-wall profiles. Often there are several of these laminated organelles per cell. A undergraduate or rotation project might entail starving and asking whether your autofluorescent signal is different from growing cells.

Response to Reviewers

Reviewer #1 (Remarks to the Author):

This is a very well written manuscript describing a state-of-the-art characterization of the red body in *Nannochloropsis* and related Eustigmatophytes. Most of the analysis is based on sophisticated microscopy and one of the key findings is that the spot changes in sized in parallel to cellcycle/cell division in a diurnal manner. Compositional analysis on isolated and partially purified red bodies is provided using fluorescence, mass spectrometry and FTIR spectroscopy to show that ketocarotenoids and long chain alcohols, possible precursors of the cell wall material algaenan, are present in the red spot. Overexpression of a cDNA encoding an enzyme involved in ketocarotenoid biosynthesis from a different alga and targeted to the plastid resulted in enlarged red bodies. Based on these data, a new hypothesis was postulated that red bodies are involved in carrying algaenan precursors from the plastid to the apoplast where they are incorporated into the cell walls of newly formed daughter cells.

Overall, this novel hypothesis on red body function is very intriguing and plausible based on the data. Most of the data are descriptive or correlation-based, and the hypothesis raises a lot of questions regarding mechanisms. For example, the data show that the red body is surrounded by a membrane but how the red body actually forms inside the plastid envelope system or how the ketocarotenoids leave the plastid remains to be seen. Some obvious follow up experiments would include proteomics of the purified red bodies to identify possible enzymes or proteins involved in red body formation to generate a more mechanistic hypothesis and targets for gene disruption to test this hypothesis. Furthermore, lipidomics of the red body would help to clarify whether neutral lipids or plastid membrane lipids are present that could provide further hints towards the origin of the red body.

- *We thank Reviewer 1 for this feedback. As suggested, we have added proteomics data for the shed red bodies, and several proteins with a putative role in red body biogenesis were identified. We agree that these data raise a lot of very intriguing questions that will be addressed in future studies also focusing on the red body lipid composition.*

Given that I do not see any technical issues with the provided analysis, the methods used, or the presentation of the data, the main question is whether there is enough novelty, general interest, and support for an intriguing new hypothesis on red bodies in Eustigmatophytes to warrant publication. My personal answer to this question is a resounding yes. General interest will probably be based on the fact that *Nannochloropsis* is an merging model system for biotechnological applications, and the intriguing biology and evolutionary history of Eustigmatophytes.

- *We thank Reviewer 1 for these positive comments.*

As a minor note, the authors may want to replace plasmid with plastid in line 104.

- *Thank you for catching this. The typo has been corrected to read, "...and the outermost plastid membrane."*

Reviewer #2 (Remarks to the Author):

I have attached below a pdf that contains my review

(text copy pasted here below from the pdf)

This is an excellent paper, with many creative experimental approaches, in-depth analyses, and outstanding writing and figures. I'm not qualified to assess the biochemical analyses and trust another reviewer is so endowed. It would be useful to learn to what extent these analyses comport with vs. diverge from those reported in Scholz et al for *gaditana*.

I'll first offer a few comments on the paper and then some unpublished data from our lab that might be useful to the authors.

Possibly the authors hit a limit in references cited, but if not, they might consider adding this: Wolf et al. (2018) *Photosynth. Res.* 135:177-189.

- *Thank you for this suggestion. We have added the following sentence and citation to the Results section regarding the initial observations of red body autofluorescence: "Our observation of autofluorescence from the red body is consistent with micrographs of a different eustigmatophyte, which was shown to be capable of using far red light for photosynthesis (citation)."*

And this should for sure be included as it has IMO standard-EM images of the Nanno wall:

Murakami R, Hashimoto H. 2009. Unusual nuclear division in *Nannochloropsis oculata* (Eustigmatophyceae, Heterokonta) which may ensure faithful transmission of secondary plas-ds. *Protist* 160:41–49. hep: [//dx.doi.org/10.1016/j.pro-s.2008.09.002](https://doi.org/10.1016/j.pro-s.2008.09.002)

- *A citation for this paper has been added to the Results section regarding the description of the red sediment in the sentence, "TEM revealed the rolled and wrinkled appearance of the autosporangial walls (Figure 6C) that were similar in thickness (~5 nm) to the outer, algaenan cell wall layer of *Nannochloropsis* (Figure 6D)."*

Line 99. Make clear here that this is max diameter observed and that evidence for smaller sizes is given below.

- *The part of the sentence describing observed diameters in TEM micrographs has been expanded into a separate sentence reading, "The maximum observed diameter was ~300-400 nm in diameter, although smaller ones were observed and will be described in later experiments."*

Line 137 should be 4C.

- *This sentence, “Green autofluorescent punctae were observed adhered to the glass after autospore separation, sometimes trapped within a transparent, pointed, tubular structure (Figure 4B).” describes an image from panel B. The issue may be in the somewhat unconventional layout of the multi-panel, multi-image figure, where the letters refer to the different time points in the experiment. Panel B describes the observation of shed cell walls with the shed red body trapped within, while panel C describes freshly released autospores, which do not have observable red body autofluorescence.*
- *A sentence has been added to the figure legend at the beginning to clarify this arrangement, “Subpanel labels A, B, and C refer to horizontal groupings consisting of a cross section schematic of the experimental procedure involving a 6-well culture plate, and accompanying micrographs to the right. “*

Line 174 IMO it is misleading to say that the maternal autosporangial wall undergoes degradation; instead, it splits open and then lasts forever!

- *This sentence has been edited to replace “degradation” with “initial splitting”, and now reads, “Together, these observations are consistent with a gap in algaenan coverage between the initial splitting of the maternal autosporangial wall and the formation of new algaenan layers around autospores.”*

Line 178 The red sediment is also reported in Scholz et al.

- *The citation has been updated to include the Scholz paper as well as the Roldolfi one about media recycling.*

Line 334 I’m surprised that an SDS gel of the protein fraction isn’t provided to give some idea as to whether it’s many faint bands or a few prominent ones.

- *Following the reviewer’s comment, we have provided an analysis of the proteins in isolated red bodies. The red body was solubilized by treatment with 5% SDS and 0.1 M DTT incubated at 95°C. SDS-PAGE revealed a band at 29 kDa together with a background smear. The solubilized proteins were subsequently subjected to proteomic MS analysis, showing that a lipocalin-like protein (NoLCN) was the most abundant peptide in the red body. These data have been added in a new Table 1, Table 1-supplemental 1 and 2, and the results are described in the main manuscript in a new section entitled: *Proteomics identifies specific candidate proteins found in isolated, shed red bodies.**

///

Unpublished data (I’m happy to provide the Niyogi lab with any full-res images they’re interested in.)

In Scholz et al we usually washed out the carotenoids, concerned that they might be contaminants. Aker publication, however, Taylor Weiss ran a carotenoid analysis of shed walls which supports your analysis. I’m not suggesting that his data be included but it’s always nice to see confirmation! We note presence of amino acids in pressed walls (Table 4), which are not expected to contain exocytosed red bodies, suggesting they’re in wall itself as well as in red body.

Our QFDE images of walls (Fig. 2 of Scholz) shows only walls released by French Press but in fact

we also looked at shed mother walls:

These show that the cellulose layer is retained, confirming your Calcoflor data. Our shed walls curl around whereas yours are straight (your Fig.6); difference might be gaditana, different suspending media (all our wall preps are in water), or something else.

To this point, our pressed walls do not curl up (Fig 2 Scholz). My interpretation is that the mother walls are released via a longitudinal slit, akin to these shells, and they then curl back on themselves. If this is the case, then presumably most of the red bodies they might have harbored would be released, explaining why they are apparently absent from the images. So how does this slit work? Here are some musings. A proto-slit might form prior to daughter cell formation, where this en-ty might account for the increase in permeability you observe. Splitting might occur in conjunction with, or subsequent to, red body release, with a portion of its contents containing an autolysin-analogue that opens up the proto-slit. Your cool idea that the red body also releases daughter-cell algaenans in that interval is totally concordant.

Here are several images that may represent the red body poised for secretion or undergoing secretion. The first two are from Santos and Leedale (Nova Hedwigia 60:219, 1995) and Suda et al (ref in your paper)

The rest are from various Nannos:

W124-03008 is particularly interesting due to its striated contents. In your video you label an organelle as a red body, but next to it is a similar organelle with striated contents:

I have numerous images with striated interiors, most in cells that were N-starved to induce TAG production. Here's one:

So this suggests a story: that when cells enter stationary via N-starve, the algaenan in their red bodies goes ahead and creates layers of outer-wall profiles. Oken there are several of these laminated organelles per cell. A undergraduate or rotation project might entail starving and asking whether your autofluorescent signal is different from growing cells.

///

Reviewer #3 (Remarks to the Author):

The presence of red bodies in Eustigmatophyceae including *Nannochloropsis* has been observed previously by other researchers. The MS "A proposed function for the red body of *Nannochloropsis* in the formation of the recalcitrant cell wall polymer, algaenan", taking advantage of the emerging model marine alga *Nannochloropsis oceanica*, performed a biochemical characterization of red bodies and proposed their function in transporting algaenan

precursors from plastid to apoplast to be incorporated into daughter cell walls. While providing well-presented biochemical data, the study lacks genetic proofs to support the conclusion.

1. Where does the red body originate from? Based on the carotenoid profiles of red body, it seemed that only ketocarotenoids were present. Why no other carotenoids like beta-carotene? Or why red body only transports ketocarotenoids out of the plastid? Where is the C32 alkyl diol synthesized?

- *Overexpression of cTP-CzBKT resulted in increased accumulation of ketocarotenoids and altered morphology of the red body, see Figure 8. No changes in ketocarotenoid accumulation was observed when CzBKT without a chloroplast targeting signal was expressed, as mentioned on line 214-215. The metabolic flux of ketocarotenoids from the chloroplast to the red body can be a result of the interaction (Line 214-224) between the chloroplast during red body formation as shown in figure 3, Suggesting that the red body originates from the chloroplast.*
- *We do not have data showing how the red body discriminates between loading of ketocarotenoids and non-ketocarotenoids, and we find this interesting question to be beyond the scope of this paper. As to why only ketocarotenoids are found in the red body we can only speculate. Interestingly, it is established that ketocarotenoids such as astaxanthin have higher antioxidant activity than e.g. beta-carotene in particular in a non-liposomal environment (Y. M. A. Naguib, Journal of Agricultural and Food Chemistry 2000, 48, 1150-1154.). These antioxidant properties of ketocarotenoids could guide future studies on the function of the red body during cell division. We have added a sentence noting the antioxidant properties of ketocarotenoids in non-liposomal conditions in 399-406.*
- *To our knowledge no biosynthetic enzymes have been shown experimentally to be involved in the biosynthesis of C32 alkyl diols, thus subcellular localization of these enzymes has not been shown. As mentioned in line 428-435, precursors of other recalcitrant biopolymers such as sporopollenin and cutin are biosynthesized in plastids, and it is plausible that C32 alkyl synthases have a similar subcellular localization.*

2. The authors collected red bodies for lipid and carotenoid profiling. These red bodies should also be used for proteomic analysis, with an aim to probe the difference compared to lipid bodies and possible proteins involved in red body biogenesis.

- *See our response to Reviewer #2 above. As requested, proteomic data of solubilized red body protein have been provided in a new Table 1, plus Table 1- supplemental 1 and 2. Results are described in the main manuscript in a new section entitled: *Proteomics identifies specific candidate proteins found in isolated, shed red bodies*. We have compared composition of red body proteins with proteins identified in *N. oceanica* lipid bodies (Vieler et al 2012), and *Phaeodactylum tricornutum* (Yonada et al. 2016 , Wang et al 2017, and Lupette et al 2017). No proteins detected in the red body were homologs of proteins in isolated lipid bodies in the previous studies.*

3. Red bodies harbor C32 alkyl diol, the possible precursor of algaenan; this can give an indication but is not solid enough. Additional evidences should be included to underpin the

conclusion. Are there ways to impair red body biogenesis (based on the proteome) ? Such mutants would be of help for the authors to confirm the role of red body in algaenan formation.

- We generated knock-out mutants of the lipocalin-like protein (NoLCN), which was shown to be the most abundant protein in the red body proteome. This is shown in Table 1, and Table 1- supplemental 1. However, no change in the morphology and fluorescent properties of the red body was observed. These data have been added in Table 1- supplemental 2. Additional studies beyond the scope of this paper will be required to determine the role of NoLCN in the red body.

Reviewer #4 (Remarks to the Author):

I was asked to evaluate the quality of the electron microscopy (tomography) data of the manuscript “A proposed function for the red body of *Nannochloropsis* in the formation of the recalcitrant cell wall polymer, algaenan” (NCOMMS-23-46997). In this context the methods used and data shown in the manuscript are excellent and state-of-the-art. Besides high pressure freezing to prepare resin embedded samples for TEM investigations the authors also used Cryo-FIB-SEM to image plunge frozen samples as close as possible to their natural state. Both methods produced excellent images of the ultrastructure of the red bodies. Specifically Cryo-FIB-SEM is technically very challenging and the images/videos produced by the authors are outstanding. The conclusions (red bodies are surrounded by a membrane, associated with the chloroplast, red bodies are shed in the apoplast, etc.) are well supported by the images and very interesting. Thus, I can conclude that the methods used and the quality of the images/data from the electron microscopical images are excellent and well suited to be published in Nature Communications. I also have extensive experience in confocal laser scanning microscopy. The methods used and the quality of the images of the light microscopical work is also excellent and suitable for publication in Nature Communication. Since I am not a specialist in the field of red bodies and algae other reviewers need to decide if the scientific value of the observations merits publication in Nature Communications. As already mentioned the quality of the methods and data of the electron and light microscopical work is definitely worth being published in Nature Communications. However, there are some major issues with the manuscript and the authors would need to address these before the manuscript can be published in Nature Communications.

Major issues:

The use of tomography throughout the manuscript that I have received is misleading and should be erased when referring to the electron microscopy studies. Tomography usually results in a 3D reconstruction of the investigated structure which is lacking. I was looking for a 3D reconstruction of a tilt series of the red body in the Cryo-TEM but could not find one even though it is described in the material and methods sections – it might have gone lost during the upload. Then I was looking of a 3D reconstruction of the red body (based on serial sections) imaged by Cryo-FIB-SEM but could not find one either. All I found was a video (10 seconds) that showed several sections (probably 50 sections or so produced with Cryo-FIB-SEM) through an

algae cell including half (not a whole) of a red body but the 3D reconstruction which is announced in the manuscript is missing. Did the authors perform 3D reconstruction of a whole/parts of a red body? If yes where is it in the manuscript? What do the authors mean by tomography? Tomography is usually a tilt series of one section (up to 300nm in thickness) in the TEM which is then used for 3D reconstruction. This is definitely not provided in the manuscript that I received. Also if the authors want to perform a 3D reconstruction of the red body (based on serial sections) then they would need to slice/mill through a whole one (currently only half of one is sliced through with Cryo-FIB-SEM) and then reconstruct it with Image J or similar software. Please clarify this issue. The data is already impressive as it is and a 3D reconstruction (even though it would be even more impressive) is not necessary as it would not really support the data any further.

- *The tilt series as well as the full reconstructed tomographic volume used for the tomogram data presented in Figure 2 C and F, Figure 2- supplemental 2, and Video 1 have been uploaded to the Electron Microscopy Data Bank (EMD-43893) and are now publicly accessible and open for review. The terminology for the main body of the text and methods has been streamlined to better highlight the methods employed, which were plunge freezing, cryo-fluorescence, cryo-FIB-SEM, and cryo-TEM. Figure 2- supplemental 2 was created to complement the initially submitted Figure 2- supplemental 1 to more fully illustrate this workflow.*

Video 1: The uploaded video 1 (10 seconds) does not show a reconstruction of a 400nm thick lamella and definitely not Cryo electron tomography. I am not sure if an incorrect file was uploaded or if the authors don't understand the term tomography. It would be nice to provide a figure based on Video 1 where the 3D reconstruction that is claimed in the figure legend of Video 1 is provided. Also, it is unclear what methods were used to produce video 1 (I assume the video shows Cryo-FIB-SEM milling). The method should be described in the figure legend of video 1.

- *See related comments and revisions in response to the previous comment. Video 1 is a movie of a reconstructed tomogram from a cryo-TEM tilt series taken of a ~350 nm thick cryo-lamella created with cryo-FIB-SEM. Video 1 was expanded to include the full reconstructed tomogram, including a complete red body. Resolution is 2.2 nm per pixel.*

Scale Bars, figure labels, image arrangement etc.: Please read the comments below, go through all of your figures, and adjust scale bars, add labels, etc. according to my comments in all images.

In Figure 1 the authors mention at the very end that the scale bar in the first image applies to all other images. It would be nice if the authors could include a similar sentence in all other figures where this applies. For example the legend of Figure 1 – S1 does not even mention the scale bar. In Figure 1- S2 it is unclear if the scale bar in the first image applies to all other images.

- *Figures 1 S1 and S2 now include the sentence "Scale bar = 5 μm and applies to all images in this figure."*

Scale bars in Figure 2 and Figure 6 are at different x,y positions (some are in the left corner some are in the right corner of the image) and different width. Please arrange them properly at similar positions (either left or right bottom corner with same distance from the bottom and side) relative to the image. I personally think that the scale bar in the 2nd image of Figure 2 of the second row is too long (almost the width of the whole image). Either way please arrange the scale bars so the figure looks more appealing.

- *The scale bars in Figure 2 have been adjusted to be of consistent thickness and placement on the images.*
- *Figure 6 scale bars and labels have also been adjusted (see comments below).*

Figure 2: Images in the second row don't have a label. Please use labels (D, E, F) in the images of the second row. Also I find the labeling for organelles (colors, lower case letters) unusual. I personally would use white or black upper case letters and label the mitochondria with "M", the nucleus with "N", the chloroplasts with "C" and the red body with "RB" throughout the manuscript (including the video). That is how it is usually done in electron micrographs. It is also unclear which images derive from resin embedded samples and which one are from Cryo-FIB-SEM. This should be clearly stated in all the figure legends. I also think that the figure legend should be expanded to clearly describe the images in more detail. The authors need to keep in mind that some readers will have very little background in (electron) microscopy and will not know which methods were used to produce the images and what they are looking at. Especially since almost all of the data in this manuscript derive from micrographs, proper figure legends are essential to make this manuscript impactful.

- *The annotations for organelles have been reformatted with colors removed, letters capitalized, and abbreviations changed as described by the reviewer. A light gray rather than stark white was chosen for the annotations to be visible but without distracting excessively from the image contents.*
- *The figure legend stated that A and B are TEM images of resin embedded cells, and that C is from cryo TEM. We believe the original layout was not clear, as the lower row is actually a higher magnification image of the same structures in the upper row, and not additional observations/cells. As suggested, to make this clearer, D, E, F subpanel labels have been added and described in the legend. Additionally, more white space has been added between A/D, B/E, C/F, and a horizontal label with "resin TEM" and "cryo TEM" has been added for additional identification.*
- *Additional details have been added to the legend to aid readers in interpreting the images and structures depicted.*

Figure 4 S1: please include scale bars.

- *The scale bar and label have been made larger and more obvious, and a note added in the figure legend to explain that it applies to all images. Additionally, dividing lines have been added to make the boundaries between images clearer, and labeling for "dividing cells" and "shed walls and red bodies" has been added to further clarify what is being shown.*

Figure 5: size of scale bar is unclear; please specify.

- *The size of the scale bar is stated in the legend, and additionally a label to the scale bar in the image has been added stating "5 μm".*

Figure 6: This is a very ugly figure and must not be published in its current form! The figure legend also lacks a basic description of the findings displayed in the images. This figure would get an F (probably 0 points) in my SEM/TEM class. Here are a few suggestions to improve the figure: please arrange scale bars in the different images in similar x,y positions (bottom right or left corner, same distance from the bottom). Please use the same font for the description of the size of the scale bars. Please arrange the description of the scale bars uniformly either on top or below of all scale bars. Please label all images with a letter (right now only a few are labeled with A, B, C, D). The abbreviation DIC is not explained in the figure legend – please explain. Please introduce some space between the images so that it is clear where one image ends and the other one starts. Please crop and arrange the images properly. Also it is unclear what the images show and which microscopical technique was used. Please include this in the image description so that the reader can understand and not guess what is depicted in the images.

- *The scale bars and scale bar labels have been made uniform in appearance at the lower left corner of each image.*
- *Each image now has a subpanel label.*
- *Images have been arranged with more white space to delineate them, and also grouped by technique (light microscopy, whole mount negative stain, resin embedded thin sections).*
- *Further detail has been added to the figure legend.*

Figure 7: Scale bar is too small and should be made larger. It will be impossible to read the size of the scale bar in the final pdf. Please include the scale bar size in the figure legend.

- *The scale bar and label have been made white for better contrast, and the text of the scale bar label enlarged. The scale bar size is now described in the figure legend as well.*

REVIEWERS' COMMENTS

Reviewer #1 (Remarks to the Author):

I briefly reviewed the revised MS and responses by all reviewers. My comments have been fully addressed especially through inclusion of new proteomics data, which were also suggested by others. I have no further suggestions for improvement of the paper.

Reviewer #2 (Remarks to the Author):

The major change made in this revision is that, as suggested by me and another reviewer, the protein composition of the red bodies was analyzed, a major polypeptide was identified as a lipocalin-like protein NoLCN, its encoding gene was inactivated, and no obvious phenotype was observed. While I'm sure this outcome was disappointing to the investigators, they did what was requested.

Therefore, I again find this a thorough, meticulous and thoughtful analysis of a new organelle. The attendant data on cell wall biochemistry, and on cell cycle-dependent properties such as the increased permeability at the time of autospore formation, will also be of value in future studies of this important organism.

I attach a pdf where I've offered a few edits, including one in the Figures.

Reviewer #3 (Remarks to the Author):

In the revised version, the authors added a couple of experiments such as proteomic analysis of RDs and knockout of an abundant RD protein. The knockout effort, however, didn't cause any observed phenotype.

While the work provides substantial information about RDs of *N. oceanica*, the proposed function of RD lacks solid proofs. Besides, there is concern about the broad interest of this work

Reviewer #4 (Remarks to the Author):

The quality of the electron microscopy (tomography) and confocal laser scanning microscopy data of the manuscript "A proposed function for the red body of *Nannochloropsis* in the formation of the recalcitrant cell wall polymer, algaenan" (NCOMMS-23-46997) is excellent and state-of-the-art. The authors have addressed all of my concerns and the manuscript should be published in "Nature Communications". Since I am not a specialist in the field of red bodies and algae other reviewers

need to decide if the scientific value of the observations merits publication in "Nature Communications". From my point of view the data merits publication in "Nature Communications".

The only minor suggestion that I have is that the authors should make sure that the scale bars and image labeling is clearly visible in the final publication. For example in Figure 5 the scale bar is barely visible and might need to be larger in the final publication. In Figure 2 the labeling for cell wall (CW) is white and hardly visible (it could be printed black for better contrast and visibility). In Figure 4 the scale bar is at the very bottom and hardly visible. In Figure 1 the scale bar is also too small and should probably be white.